# `LeadCache`: Regret-Optimal Caching in Networks

**Debjit Paria**[*]
Department of Computer Science
Chennai Mathematical Institute
Chennai 603103, India
debjit.paria1999@gmail.com

**Abhishek Sinha**
Department of Electrical Engineering
Indian Institute of Technology Madras
Chennai 600036, India
abhishek.sinha@ee.iitm.ac.in

## Abstract

We consider an online prediction problem in the context of network caching. Assume that multiple users are connected to several caches via a bipartite network. At any time slot, each user may request an arbitrary file chosen from a large catalog. A user's request at a slot is met if the requested file is cached in at least one of the caches connected to the user. Our objective is to predict, prefetch, and optimally distribute the files on the caches at each slot to maximize the total number of cache hits. The problem is non-trivial due to the non-convex and non-smooth nature of the objective function. In this paper, we propose `LeadCache` - an efficient online caching policy based on the Follow-the-Perturbed-Leader paradigm. We show that `LeadCache` is regret-optimal up to a factor of $\tilde{O}(n^{3/8})$, where $n$ is the number of users. We design two efficient implementations of the `LeadCache` policy, one based on Pipage rounding and the other based on Madow's sampling, each of which makes precisely one call to an LP-solver per iteration. Furthermore, with a Strong-Law-type assumption, we show that the total number of file fetches under `LeadCache` remains almost surely finite over an infinite horizon. Finally, we derive an approximately tight regret lower bound using results from graph coloring. We conclude that the learning-based `LeadCache` policy decisively outperforms the state-of-the-art caching policies both theoretically and empirically.

## 1 Introduction

We consider an online structured learning problem, called Bipartite Caching, that lies at the core of many large-scale internet services, including Content Distribution Networks (CDN) and Cloud Computing. Formally, a set $\mathcal{I}$ of $n$ users is connected to a set $\mathcal{J}$ of $m$ caches via a bipartite network $G(\mathcal{I} \uplus \mathcal{J}, E)$. Each cache is connected to at most $d$ users, and each user is connected to at most $\Delta$ caches (see Figure 1 (b)). There is a catalog consisting of $N$ unique files, and each of the $m$ caches can host at most $C$ files at a time (in practice, $C \ll N$). The system evolves in discrete time slots. Each of the $n$ users may request any file from the catalog at each time slot. The file requests could be dictated by an adversary. Given the storage capacity constraints, an online caching policy decides the files to be cached on different caches at each slot before the requests for that slot arrive. The objective is to maximize the total number of *hits* by the unknown incoming requests by coordinating the caching decisions among multiple caches in an online fashion. The Bipartite Caching problem is a strict generalization of the online $k$-**sets** problem that predicts a set of $k$ items at each round so that the predicted set includes the item chosen by the adversary [Koolen et al., 2010, Cohen and Hazan, 2015]. However, unlike the $k$-**sets** problem, which predicts a single subset at a time, in this problem, we are interested in sequentially predicting multiple subsets, each corresponding to one of the caches. The interaction among the caches through the non-linear reward function makes this problem challenging.

---

[*]Work done at the Indian Institute of Technology Madras as a part of the first author's Master's thesis.

35th Conference on Neural Information Processing Systems (NeurIPS 2021).

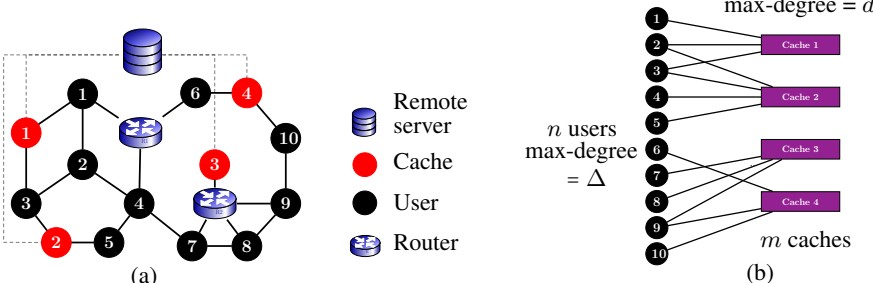

Figure 1: Reduction of the Network Caching problem (a) to the Bipartite Caching problem (b). In this schematic, we assumed that a cache, located within two hops, is reachable to a user.

The Bipartite Caching problem is a simplified abstraction of the more general Network Caching problem central to the commercial CDNs, such as Akamai [Nygren et al., 2010], Amazon Web Services (AWS), and Microsoft Azure [Paschos et al., 2020]. In the Network Caching problem, one is given an arbitrary graph $\mathcal{G}(V, E)$, a set of users $\mathcal{I} \subseteq V$, and a set of caches $\mathcal{J} \subseteq V$. A user can retrieve a file from a cache only if the cache hosts the requested file. If the $i^{\text{th}}$ user retrieves the requested file from the $j^{\text{th}}$ cache, the user receives a reward of $r_{ij} \geq 0$ for that slot. If the requested file is not hosted in any of the caches reachable to the user, the user receives zero rewards for that slot. The goal of a network caching policy is to dynamically place files on the caches so that cumulative reward obtained by all users is maximized. The Network Caching problem reduces to the Bipartite Caching problem when the rewards are restricted to the set $\{0, 1\}$. It will be clear from the sequel that the algorithms presented in this paper can be extended to the general Network Caching problem as well.

## 1.1 Problem Formulation

Denote the file requested by the $i^{\text{th}}$ user by the one-hot encoded $N$-dimensional vector $\boldsymbol{x}_t^i$. In other words, $\boldsymbol{x}_{tf}^i = 1$ if the $i^{\text{th}}$ user requests file $f \in [N]$ at time slot $t$, or $\boldsymbol{x}_{tf}^i = 0$ otherwise. Since a user may request at most one file per time slot, we have: $\sum_{f=1}^N \boldsymbol{x}_{tf}^i \leq 1, \quad \forall i \in \mathcal{I}, \forall t$. An online caching policy prefetches files on the caches at every time slot based on past requests. Unlike classical caching policies, such as LRU, LFU, FIFO, Marker, that fetch a file immediately upon a cache-miss, we do not enforce this constraint in the problem statement. The set of files placed on the $j^{\text{th}}$ cache at time $t$ is represented by the $N$-dimensional incidence vector $\boldsymbol{y}_t^j \in \{0, 1\}^N$. In other words, $\boldsymbol{y}_{tf}^j = 1$ if the $j^{\text{th}}$ cache hosts file $f \in [N]$ at time $t$, or $\boldsymbol{y}_{tf}^j = 0$ otherwise. Due to cache capacity constraints, the following inequality must be satisfied at each time slot $t$: $\sum_{f=1}^N \boldsymbol{y}_{tf}^j \leq C, \quad \forall j \in \mathcal{J}$.

The set of all admissible caching configurations, denoted by $\mathcal{Y} \subseteq \{0, 1\}^{Nm}$, is dictated by the cache capacity constraints. In principle, the caching policy is allowed to replace all elements of the caches at every slot, incurring a potentially huge downloading cost over an interval. However, in Section 4, we show that the total number of files fetched to the caches under the proposed LeadCache policy remains almost surely finite under very mild assumptions on the file request process.

The $i^{\text{th}}$ user receives a *cache hit* at time slot $t$ if and only if *any* of the caches connected to the $i^{\text{th}}$ user hosts the file requested by the user at slot $t$. In the case of a cache hit, the user obtains a unit reward. On the other hand, in the case of a *cache miss*, the user receives zero rewards for that slot. Hence, for a given aggregate request vector from all users $\boldsymbol{x}_t = (\boldsymbol{x}_t^i, i \in \mathcal{I})$ and the aggregate cache configuration vector of all caches $\boldsymbol{y}_t = (\boldsymbol{y}_t^j, j \in \mathcal{J})$, the total reward $q(\boldsymbol{x}_t, \boldsymbol{y}_t)$ obtained by the users at time $t$ may be expressed as follows:

$$q(\boldsymbol{x}_t, \boldsymbol{y}_t) \equiv \sum_{i \in \mathcal{I}} \boldsymbol{x}_t^i \cdot \min\left\{\boldsymbol{1}_{N \times 1}, \left(\sum_{j \in \partial^+(i)} \boldsymbol{y}_t^j\right)\right\}, \tag{1}$$

where $\boldsymbol{a} \cdot \boldsymbol{b}$ denotes the inner-product of the vectors $\boldsymbol{a}$ and $\boldsymbol{b}$, $\boldsymbol{1}_{N \times 1}$ denotes the $N$-dimensional all-one column vector, the set $\partial^+(i)$ denotes the set of all caches connected to the $i^{\text{th}}$ user, and the "min" operator is applied component wise. The total reward $Q(T)$ accrued in a time-horizon of length $T$

is obtained by summing the slot-wise rewards, *i.e.*, $Q(T) = \sum_{t=1}^{T} q(\boldsymbol{x}_t, \boldsymbol{y}_t)$. Following the standard practice in the online learning literature, we measure the performance of any online policy $\pi$ using the notion of (static) *regret* $R^\pi(T)$, defined as the maximum difference in the cumulative rewards obtained by the optimal fixed caching-configuration in hindsight and that of the online policy $\pi$, *i.e.*,

$$R^\pi(T) \stackrel{\text{(def.)}}{=} \sup_{\{\boldsymbol{x}_t\}_{t=1}^T} \left( \sum_{t=1}^{T} q(\boldsymbol{x}_t, \boldsymbol{y}^*) - \sum_{t=1}^{T} q(\boldsymbol{x}_t, \boldsymbol{y}_t^\pi) \right), \tag{2}$$

where $\boldsymbol{y}^*$ is the best static cache-configuration in *hindsight* for the file request sequence $\{\boldsymbol{x}_t\}_{t=1}^T$, *i.e.*, $\boldsymbol{y}^* = \arg\max_{\boldsymbol{y} \in \mathcal{Y}} \sum_{t=1}^{T} q(\boldsymbol{x}_t, \boldsymbol{y})$. We assume that the file request sequence is generated by an *oblivious adversary*, *i.e.*, the entire request sequence $\{\boldsymbol{x}_t\}_{t \geq 1}$ is fixed a priori. Note that the problem is *non-convex*, as we seek binary cache allocations. With an eye towards efficient implementation, later we will also consider the problem of designing efficient policies that guarantee a sub-linear $\alpha$-regret for a suitable value of $\alpha < 1$ [Garber, 2021, Kakade et al., 2009, Fujita et al., 2013].

## 2   Background and Related Work

Online Linear Optimization (OLO) is a canonical online learning problem that can be formulated as a repeated game played between a learner (also known as the forecaster) and an adversary [Cesa-Bianchi and Lugosi, 2006]. In this model, at every time slot $t$, the policy selects an action $\boldsymbol{y}_t$ from a feasible set $\mathcal{Y} \subseteq \mathbb{R}^d$. After that, the adversary reveals a reward vector $\boldsymbol{x}_t$ from a set $\mathcal{X} \subseteq \mathbb{R}^d$. The adversary is assumed to be *oblivious*, *i.e.*, the sequence of reward vectors is fixed before the game begins. With the above choices, the policy receives a scalar reward $q(\boldsymbol{x}_t, \boldsymbol{y}_t) := \langle \boldsymbol{x}_t, \boldsymbol{y}_t \rangle$ at slot $t$. A classic objective in this setting is to design a policy with a small regret. Follow the Perturbed Leader (FTPL), is a well-known online policy for the OLO problem [Hannan, 1957]. At time slot $t$, the FTPL policy adds a random noise vector $\boldsymbol{\gamma}_t$ to the cumulative reward vector $\boldsymbol{X}_t = \sum_{\tau=1}^{t-1} \boldsymbol{x}_\tau$, and then selects the best action against this perturbed reward, *i.e.*, $\boldsymbol{y}_t := \arg\max_{\boldsymbol{y} \in \mathcal{Y}} \langle \boldsymbol{X}_t + \boldsymbol{\gamma}_t, \boldsymbol{y} \rangle$. See Abernethy et al. [2016] for a unifying treatment of the FTPL policies through the lens of stochastic smoothing.

A large number of papers on caching assume some stochastic model for the file request sequence, *e.g.*, Independent Reference Model (IRM) and Shot Noise Model (SNM) [Traverso et al., 2013]. Classic page replacement algorithms, such as MIN, LRU, LFU, and FIFO, are designed to minimize the *competitive ratio* with adversarial requests [Borodin and El-Yaniv, 2005, Van Roy, 2007, Lee et al., 1999, Dan and Towsley, 1990]. These algorithms, being *non-prefetching* in nature, replace a page on demand upon a cache-miss. However, since the competitive ratio metric is multiplicative in nature, there can be a large gap between the hit ratio of a competitively optimal policy and the optimal offline policy. To design better algorithms, the caching problem has recently been investigated through the lens of regret minimization with *prefetching* policies that *learn* from the past request sequence [Vitter and Krishnan, 1996, Krishnan and Vitter, 1998]. Daniely and Mansour [2019] considered the problem of minimizing the regret plus the switching cost for a single cache. The authors proposed a variant of the celebrated exponential weight algorithm [Littlestone and Warmuth, 1994, Freund and Schapire, 1997] that ensures the minimum competitive ratio and a small but sub-optimal regret. The Bipartite Caching model was first proposed in a pioneering paper by Shanmugam et al. [2013], where they considered a stochastic version of the problem with known file popularities. Paschos et al. [2019] proposed an Online Gradient Ascent (OGA)-based Bipartite Caching policy that allows caching a fraction of the Maximum Distance Separable (MDS)-coded files. Closely related to this paper is the recent work by Bhattacharjee et al. [2020], where the authors designed a regret-optimal single-cache policy and a Bipartite Caching policy for fountain-coded files. However, the fundamental problem of designing a regret-optimal uncoded caching policy for the Bipartite Caching problem was left as an open problem.

**Why standard approaches fail:**   A straightforward way to formulate the Bipartite Caching problem is to pose it as an instance of the classic Prediction with Expert Advice problem [Cesa-Bianchi and Lugosi, 2006], where each of the possible $\binom{N}{C}^m$ cache configurations is treated as experts. However, this approach is computationally infeasible due to the massive number of resulting experts. Furthermore, the expected FTPL policy for convex losses in Hazan [2019] and its sampled version in Hazan and Minasyan [2020] are both computationally intensive. In view of the above challenges, we now present our main technical contributions.

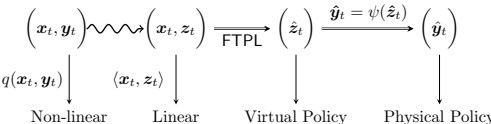

Figure 2: Reduction pipeline illustrating the translation of the Bipartite Caching problem with a non-linear reward function to an online learning problem with a linear reward function.

## 3 Main Results

**1. FTPL for a non-linear non-convex problem:** We propose LeadCache, a network caching policy based on the Follow the Perturbed Leader paradigm. The *non-linearity* of the reward function and the non-convexity of the feasible set (due to the integrality of cache allocations) pose a significant challenge in using the generic FTPL framework [Abernethy et al., 2016]. To circumvent this difficulty, we switch to a *virtual action* domain $\mathcal{Z}$ where the reward function is linear. We use an anytime version of the FTPL policy for designing a virtual policy for the linearized virtual learning problem. Finally, we translate the virtual policy back to the original action domain $\mathcal{Y}$ with the help of a mapping $\psi : \mathcal{Z} \to \mathcal{Y}$, obtained by solving a combinatorial optimization problem (see Figure 2 for the overall reduction pipeline).

**2. New Rounding Techniques and $\alpha$-regret:** The mapping $\psi$, which translates the virtual actions to physical caching actions in the above scheme, turns out to be an NP-hard Integer Linear Program. As our second contribution, we design a linear-time Pipage rounding technique for this problem [Ageev and Sviridenko, 2004]. Incidentally, our rounding process substantially improves upon a previous rounding scheme proposed by Shanmugam et al. [2013] in the context of caching with i.i.d. requests. Next, we propose a linear-time randomized rounding scheme that yields an efficient online policy with a provable sub-linear $\alpha \equiv 1 - 1/e$ regret. The proposed randomized rounding algorithm exploits a classical sampling technique used in statistical surveys.

**3. Bounding the Switching Cost:** As our third contribution, we show that if the file requests are generated by a stochastic process satisfying a mild Strong Law-type property, then the caching configuration under the LeadCache policy converges to the corresponding optimal configuration *almost surely in finite time*. As new file fetches to the caches from the remote server consume bandwidth, this result implies that the proposed policy offers the best of both worlds - (1) a sub-linear regret for adversarial requests, and (2) finite downloads for "stochastically regular" requests.

**4. New Regret Lower Bound:** As our final contribution, we derive minimax regret lower bound that is tight up to a factor of $\tilde{O}(n^{3/8})$. Our lower bound sharpens a result in Bhattacharjee et al. [2020]. The proof of the lower bound critically utilizes graph coloring theory and the probabilistic Balls-into-Bins framework.

## 4 The LeadCache Policy

In this section, we propose LeadCache - an efficient network caching policy that guarantees near-optimal regret. Since the reward function (1) is non-linear, we *linearize* the problem by switching to a *virtual* action domain $\mathcal{Z}$, as detailed below.

**The Virtual Caching Problem:** First, we consider an associated Online Linear Optimization (OLO) problem, called Virtual Caching, as defined next. In this problem, at each slot $t$, a virtual action $z_t \equiv (z_t^i, i \in \mathcal{I})$ is taken in response to the file requests received so far. The $i^{\text{th}}$ component of the virtual action, denoted by $z_t^i \in \{0,1\}^N$, roughly indicates the availability of the files in the caches connected to the $i^{\text{th}}$ user. The set of all admissible virtual actions, denoted by $\mathcal{Z} \subseteq \{0,1\}^{N \times n}$, is defined below in Eqn. (4). The reward $r(x_t, z_t)$ accrued by the virtual action $z_t$ for the file request vector $x_t$ at the $t^{\text{th}}$ slot is given by their inner product, *i.e.*,

$$r(x_t, z_t) := \langle x_t, z_t \rangle = \sum_{i \in \mathcal{I}} x_t^i \cdot z_t^i. \tag{3}$$

**Virtual Actions:** The set $\mathcal{Z}$ of all admissible virtual actions is defined as the set of all binary vectors $\boldsymbol{z} \in \{0,1\}^{N \times n}$ such that the following component wise inequalities hold for some admissible physical cache configuration vector $\boldsymbol{y} \in \mathcal{Y}$ :

$$\boldsymbol{z}^i \leq \min\left\{\mathbf{1}_{N \times 1}, \left(\sum_{j \in \partial^+(i)} \boldsymbol{y}^j\right)\right\}, \;\; 1 \leq i \leq n. \tag{4}$$

More explicitly, the set $\mathcal{Z}$ can be characterized as the set of all binary vectors $\boldsymbol{z} \in \{0,1\}^{N \times n}$ satisfying the following constraints for some feasible $\boldsymbol{y} \in \mathcal{Y}$:

$$z_f^i \;\; \leq \;\; \sum_{j \in \partial^+(i)} y_f^j, \;\; \forall i \in \mathcal{I}, f \in [N] \tag{5}$$

$$\sum_{f=1}^N y_f^j \;\; \leq \;\; C, \;\; \forall j \in \mathcal{J}, \tag{6}$$

$$y_f^j, z_f^i \;\; \in \;\; \{0,1\}, \;\; \forall i \in \mathcal{I}, \forall j \in \mathcal{J}, f \in [N]. \tag{7}$$

Let $\psi : \mathcal{Z} \to \mathcal{Y}$ be a mapping that maps any admissible virtual action $\boldsymbol{z} \in \mathcal{Z}$ to a corresponding physical caching action $\boldsymbol{y}$ satisfying the condition (4). Hence, the binary variable $z_{tf}^i = 1$ *only if* the file $f$ is hosted in one of the caches connected to the $i^{\text{th}}$ user at time $t$ in the physical configuration $\boldsymbol{y}_t = \psi(\boldsymbol{z}_t)$. The mapping $\psi$ may be used to translate any virtual caching policy $\pi^{\text{virtual}} = \{\boldsymbol{z}_t\}_{t \geq 1}$, to a physical caching policy $\pi^{\text{phy}} \equiv \psi(\pi^{\text{virtual}}) = \{\boldsymbol{y}_t\}_{t \geq 1}$ through the correspondence $\boldsymbol{y}_t = \psi(\boldsymbol{z}_t), \forall t \geq 1$. The following lemma relates the regrets incurred by these two online policies:

**Lemma 1.** *For any virtual caching policy $\pi^{virtual}$, define a physical caching policy $\pi^{phy} = \psi(\pi^{virtual})$ as above. Then the regret of the policy $\pi^{phy}$ is bounded above by that of the policy $\pi^{virtual}$, i.e.,*

$$R_T^{\pi^{phy}} \leq R_T^{\pi^{virtual}}, \;\; \forall T \geq 1.$$

Please refer to Section 10.1 in the supplementary material for the proof of Lemma 1. Lemma 1 implies that any low-regret virtual caching policy may be used to design a low-regret physical caching policy using the non-linear mapping $\psi(\cdot)$. The key advantage of the virtual caching problem is that it is a standard OLO problem. Hence, in our proposed `LeadCache` policy, we use an anytime version of the `FTPL` policy for solving virtual caching problem. The overall `LeadCache` policy is described below in Algorithm 1:

---

**Algorithm 1** The `LeadCache` Policy
---
1: $\boldsymbol{X}(0) \leftarrow \boldsymbol{0}$
2: Sample $\boldsymbol{\gamma} \overset{\text{i.i.d.}}{\sim} \mathcal{N}(0, \mathbf{1}_{Nn \times 1})$
3: **for** $t = 1$ to $T$ **do**
4: $\quad \boldsymbol{X}(t) \leftarrow \boldsymbol{X}(t-1) + \boldsymbol{x}_t$
5: $\quad \eta_t \leftarrow \dfrac{n^{3/4}}{(2d(\log \frac{N}{C}+1))^{1/4}} \sqrt{\dfrac{t}{Cm}}$
6: $\quad \boldsymbol{\Theta}(t) \leftarrow \boldsymbol{X}(t) + \eta_t \boldsymbol{\gamma}$
7: $\quad \boldsymbol{z}_t \leftarrow \max_{\boldsymbol{z} \in \mathcal{Z}} \langle \boldsymbol{\Theta}(t), \boldsymbol{z} \rangle.$
8: $\quad \boldsymbol{y}_t \leftarrow \psi(\boldsymbol{z}_t).$
9: **end for**

---

In Algorithm 1, the flattened $Nn \times 1$ dimensional vector $\boldsymbol{X}(t)$ denotes the cumulative count of the file requests (for each (user, file) tuple), and the vector $\boldsymbol{\Theta}(t) = (\boldsymbol{\theta}^i(t), 1 \leq i \leq n)$ denotes the perturbed cumulative file request counts obtained upon adding a scaled i.i.d. Gaussian noise vector to $\boldsymbol{X}(t)$. It is also possible to sample a fresh Gaussian vector $\boldsymbol{\gamma}_t$ in step 6 at every time slot, leading to a high probability regret bound [Devroye et al., 2015]. The following Theorem gives an upper bound on the regret achieved by `LeadCache`:

**Theorem 1.** *The expected regret of the `LeadCache` policy is upper bounded as:*

$$\mathbb{E}(R_T^{\textit{LeadCache}}) \leq \kappa n^{3/4} d^{1/4} \sqrt{mCT},$$

*where $\kappa = O(\textsf{poly-log}(N/C))$, and the expectation is taken with respect to the random noise added by the policy.*

Note that in contrast with the generic regret bound of Suggala and Netrapalli [2020] for non-convex problems, our regret-bound has only logarithmic dependence on the ambient dimension $N$.

**Proof outline:** Our analysis of the LeadCache policy uses the elegant stochastic smoothing framework developed in Abernethy et al. [2016, 2014], Cohen and Hazan [2015], Lee [2018]. See Section 10.2 of the supplementary material for the proof of Theorem 1.

## 4.1 Fast approximate implementation

The computationally intensive procedures in Algorithm 1 are (I) solving the optimization problem in step 7 to determine the virtual caching actions and (II) translating the virtual actions back to the physical caching actions in step (8). Since the perturbed vector $\boldsymbol{\Theta}(t)$ is obtained by adding white Gaussian noise to the cumulative request vector $\boldsymbol{X}(t)$, some of its components could be negative. For maximizing the objective (7), it is clear that if some coefficient $\theta_f^i(t)$ is negative for some $(i, f)$ tuple, it is feasible and optimal to set the virtual action variable $z_f^i$ to zero. Hence, steps (7) and (8) of Algorithm 1 may be combined as:

$$\boldsymbol{y}(t) \leftarrow \arg\max_{\boldsymbol{y} \in \mathcal{Y}} \underbrace{\sum_{i \in \mathcal{I}, f \in [N]} (\theta_f^i(t))^+ \Big( \min\big(1, \sum_{j \in \partial^+(i)} y_f^j\big) \Big)}_{L(\boldsymbol{y})}, \tag{8}$$

where $x^+ \equiv \max(0, x)$. Incidentally, we find that problem (8) is mathematically identical to the uncoded *Femtocaching* problem with known file request probabilities studied by Shanmugam et al. [2013, Section III]. In the same paper, the authors proved the problem (8) to be **NP-Hard**. The authors also proposed a complex iterative rounding method for the LP relaxation of the problem, where, in each iteration, one needs to compute certain matchings. We now propose two simple *linear-time* rounding techniques that enjoy the same approximation guarantee.

**LP Relaxation:** We now introduce a new set of variables $z_f^i := \min(1, \sum_{j \in \partial^+(i)} y_f^j), \forall i, f$, and relax the integrality constraints to arrive at the following LP:

$$\max \sum_{i,f} (\theta_f^i(t))^+ z_f^i, \tag{9}$$

Subject to,

$$z_f^i \leq \sum_{j \in \partial^+(i)} y_f^j, \quad \forall i \in \mathcal{I}, f \in [N] \tag{10}$$

$$\sum_{f=1}^N y_f^j \leq C, \quad \forall j \in \mathcal{J}, \tag{11}$$

$$0 \leq y_f^j \leq 1, \quad \forall j \in \mathcal{J}, f \in [N]; 0 \leq z_f^i \leq 1, \quad \forall i \in \mathcal{I}, f \in [N]. \tag{12}$$

Denote the objective function for the problem (8) by $L(\boldsymbol{y})$ and its optimal value (over $\mathcal{Z}$) by OPT. Let $\boldsymbol{y}^*$ be an optimal solution to the relaxed LP (9) and $\mathcal{Z}_{\text{rel}}$ be the corresponding relaxed feasible set. Since LP (9) is a relaxation to (8), it naturally holds that $L(\boldsymbol{y}^*) \geq \text{OPT}$. To round the resulting cache allocation vector $\boldsymbol{y}^*$ to an integral one, we consider the following two rounding schemes - (1) Deterministic Pipage rounding and (2) Randomized sampling-based rounding.

**1. Pipage Rounding:** The general framework of *Pipage rounding* was introduced by Ageev and Sviridenko [2004]. Our rounding technique, given in Algorithm 2, is markedly simpler compared to Algorithm 1 of Shanmugam et al. [2013]. While we round two fractional allocations of a single cache at a time, the rounding procedure of Shanmugam et al. [2013] jointly rounds several allocations in multiple caches at the same time by computing matchings in a bipartite graph.

**Design:** The key to our deterministic rounding procedure is to consider the following surrogate objective function $\phi(\boldsymbol{y})$ instead of the original objective $L(\boldsymbol{y})$ as given in Eqn. (8):

$$\phi(\boldsymbol{y}) \equiv \sum_{i,f} (\theta_f^i(t))^+ \Big( 1 - \prod_{j \in \partial^+(i)} (1 - y_f^j) \Big). \tag{15}$$

Following a standard algebraic argument [Ageev and Sviridenko, 2004, Eqn. (16)], we have:

$$L(\boldsymbol{y}) \overset{(a)}{\geq} \phi(\boldsymbol{y}) \geq \Big( 1 - (1 - \frac{1}{\Delta})^\Delta \Big) L(\boldsymbol{y}), \quad \forall \boldsymbol{y} \in [0, 1]^N, \tag{16}$$

---

**Algorithm 2** Cache-wise Deterministic Pipage rounding

---

1: $\boldsymbol{y} \leftarrow$ Solution of the LP (9).
2: **while** $\boldsymbol{y}$ is not integral **do**
3:     Select a cache $j$ with two fractional variables $y_{f_1}^j$ and $y_{f_2}^j$.
4:     Set $\epsilon_1 \leftarrow \min(y_{f_1}^j, 1 - y_{f_2}^j), \epsilon_2 \leftarrow \min(1 - y_{f_1}^j, y_{f_2}^j)$.
5:     Define two new feasible cache-allocation vectors $\boldsymbol{\alpha}, \boldsymbol{\beta}$ as follows:

$$\alpha_{f_1}^j \leftarrow y_{f_1}^j - \epsilon_1, \alpha_{f_2}^j \leftarrow y_{f_2}^j + \epsilon_1, \quad \text{and} \quad \alpha_f^k \leftarrow y_f^k, \text{ otherwise,} \tag{13}$$

$$\beta_{f_1}^j \leftarrow y_{f_1}^j + \epsilon_2, \beta_{f_2}^j \leftarrow y_{f_2}^j - \epsilon_2, \quad \text{and} \quad \beta_f^k \leftarrow y_f^k, \text{ otherwise.} \tag{14}$$

6:     Set $\boldsymbol{y} \leftarrow \arg\max_{\boldsymbol{x} \in \{\boldsymbol{\alpha}, \boldsymbol{\beta}\}} \phi(\boldsymbol{x})$.
7: **end while**
8: **return** $\boldsymbol{y}$.

---

where $\Delta \equiv \max_{i \in \mathcal{I}} |\partial^+(i)|$. Note that inequality (a) holds with equality for all binary vectors $\boldsymbol{y} \in \{0, 1\}^{mN}$. Our Pipage rounding procedure, given in Algorithm 2, begins with an optimal solution of the LP (9). Then it iteratively perturbs two fractional variables (if any) in a single cache in such a way that the value of the surrogate objective function $\phi(\boldsymbol{y})$ never decreases while at least one of the two fractional variables is rounded to an integer. Step (4) ensures that the feasibility is maintained at every step of the roundings. Upon termination (which occurs within $O(mN)$ steps), the rounding procedure yields a feasible integral allocation vector $\hat{\boldsymbol{y}}$ with an objective value $L(\hat{\boldsymbol{y}})$, which is within a factor of $1 - (1 - \frac{1}{\Delta})^\Delta$ of the optimum objective. The following theorem formalizes this claim.

**Theorem 2.** *Algorithm 2 is an $\alpha = 1 - (1 - \frac{1}{\Delta})^\Delta$ approximation algorithm for the problem 8.*

See Section 11 of the supplementary material for the proof of Theorem 2. Note that the Pipage rounding procedure, although effective in practice, is not known to have a formal regret bound.

**2. An efficient policy for achieving a sub-linear $\alpha$-regret:** Since the offline problem (8) is **NP-Hard**, a natural follow-up problem is to design a policy with a sub-linear $\alpha$-regret. Recall that $\alpha$-regret is defined similarly as the usual static regret where the reward accrued by the offline oracle policy (*i.e.,* the first term in Eqn. (2)) is discounted by a factor of $\alpha$ [Kalai and Vempala, 2005]. Note that directly using the Pipage-rounded solution from Algorithm 2 does not necessarily yield an online policy with a sub-linear $\alpha$-regret [Kakade et al., 2009, Garber, 2021, Hazan et al., 2018, Fujita et al., 2013]. In the following, we give an efficient offline-to-online reduction that makes only a *single* query to a linear-time randomized rounding procedure per iteration. In our reduction, the following notion of an $\alpha$ point-wise approximation algorithm [Kalai and Vempala, 2005] plays a pivotal role.

**Definition 1** ($\alpha$ point-wise approximation). *For a feasible set $\mathcal{Z}$ in the non-negative orthant and a non-negative input vector $\boldsymbol{x}$, consider the Integer Linear Program $\max_{\boldsymbol{z} \in \mathcal{Z}} \boldsymbol{z} \cdot \boldsymbol{x}$. Let $\mathcal{Z}_{rel} \supseteq \mathcal{Z}$ be a relaxation of the feasible set $\mathcal{Z}$ and let $\boldsymbol{z} \in \arg\max_{\boldsymbol{z} \in \mathcal{Z}_{rel}} \boldsymbol{z} \cdot \boldsymbol{x}$ be an optimal solution of the relaxed ILP. If for some $\alpha > 0$, a (randomized) rounding algorithm $A$ returns a feasible solution $\hat{\boldsymbol{z}} \in \mathcal{Z}$ such that $\mathbb{E}\hat{z}_i \geq \alpha z_i, \forall i$ and for any input $\boldsymbol{x}$, we call the algorithm $A$ an $\alpha$ point-wise approximation.*

It immediately follows that for any $\alpha$ point-wise approximation algorithm for the problem 9, if $\boldsymbol{z}_t \in \mathcal{Z}_{rel}$ be the relaxed virtual action at time $t$, we have $\sum_{t=1}^T \mathbb{E}[\hat{\boldsymbol{z}}_t] \cdot \boldsymbol{x}_t \geq \alpha \sum_{t=1}^T \boldsymbol{z}_t \cdot \boldsymbol{x}_t$, where the inequality follows from the point-wise approximation property. Thus, for any $\boldsymbol{z}^* \in \mathcal{Z}$, the $\alpha$-regret of the virtual policy may be upper bounded as

$$\alpha \sum_{t=1}^T \boldsymbol{x}_t \cdot \boldsymbol{z}^* - \sum_{t=1}^T \mathbb{E}[\hat{\boldsymbol{z}}_t] \cdot \boldsymbol{x}_t \leq \alpha \Big( \sum_{t=1}^T \boldsymbol{x}_t \cdot \boldsymbol{z}^* - \sum_{t=1}^T \boldsymbol{x}_t \cdot \boldsymbol{z}_t \Big) \overset{(b)}{\leq} \alpha \mathbb{E}(\tilde{R}_T^{\texttt{LeadCache}}), \tag{17}$$

where $\tilde{R}_T^{\texttt{LeadCache}}$ is an upper bound to the regret of the `LeadCache` policy with the relaxed actions given by the solution of the LP (9). We bound the quantity $\mathbb{E}(\tilde{R}_T^{\texttt{LeadCache}})$ in the following Proposition.

**Proposition 1.** *For an appropriate learning rate sequence $\{\eta_t\}_{t \geq 1}$, the expected regret of the Lead-Cache policy with the relaxed action set $\mathcal{Z}_{rel}$ can be bounded as follows:*

$$\mathbb{E}(\tilde{R}_T^{\texttt{LeadCache}}) \leq \kappa_1 n^{3/4} \sqrt{dmCT},$$

*where $\kappa_1$ is poly-logarithmic in $N$ and $n$.*

See Section 13 of the supplementary materials for a proof sketch of the above result. In the following, we design an $\alpha$ point-wise approximate randomized rounding scheme.

**Randomized Rounding via Madow's sampling:**   The key ingredient to our $\alpha$ point-wise approximation oracle is Madow's systematic sampling scheme taken from the statistical sampling literature [Madow et al., 1949]. For a set of *feasible* inclusion probabilities $\boldsymbol{p}$ on a set of items $[N]$, Madow's scheme outputs a subset $S$ of size $C$ such that the $i^{\text{th}}$ element is included in the subset $S$ with probability $p_i, 1 \le i \le N$. For this sampling scheme to work, it is necessary and sufficient that the inclusion probability vector $\boldsymbol{p}$ satisfies the following feasibility constraint:

$$\sum_{i=1}^{N} p_i = C, \text{ and } 0 \le p_i \le 1, \forall i \in [N]. \tag{18}$$

The pseudocode for Madow's sampling is given in Section 12 in the supplement. Our proposed $\alpha$ point-wise approximate rounding scheme independently samples $C$ files in each cache in accordance with the inclusion probability given by the fractional allocation vector $\boldsymbol{y}$ obtained from the solution of the relaxed LP (9). From the constraints of the LP, it immediately follows that the inclusion vector $\boldsymbol{y}^j$ satisfies the feasibility constraint (18); hence the above process is sound. The overall rounding scheme is summarized in Algorithm 3. To show that the resulting rounding scheme satisfies the $\alpha$ point-wise approximation property, note that for all $i \in \mathcal{I}$ and $f \in [N]$:

$$\mathbb{P}(\hat{z}_f^i = 1) = \mathbb{P}(\bigvee_{j \in \partial^+(i)} \hat{y}_f^j = 1) \overset{(a)}{=} 1 - \prod_{j \in \partial^+(i)} (1 - y_f^j) \overset{(b)}{\ge} 1 - e^{-\sum_{j \in \partial^+(i)} y_f^j} \overset{(c)}{\ge} 1 - e^{-z_f^i} \overset{(d)}{\ge} (1 - \frac{1}{e}) z_f^i,$$

where the equality (a) follows from the fact that rounding in each caches are done independently of each other, the inequality (b) follows from the standard inequality $\exp(x) \ge 1 + x, \forall x \in \mathbb{R}$, the inequality (c) follows from the feasibility constraint (10) of the LP, and finally the inequality (d) follows from the concavity of the function $1 - \exp(-x)$ and the fact that $0 \le z_f^i \le 1$. Since $\hat{z}_f^i$ is binary, it immediately follows that the randomized rounding scheme in Algorithm 3 is an $\alpha$ point-wise approximation with $\alpha = 1 - 1/e$. We formally state the result in the following Theorem.

**Theorem 3.** *The* `LeadCache` *policy, in conjunction with the randomized rounding scheme with Madow's sampling (Algorithm 3), achieves an* $\alpha = 1 - e^{-1}$*-regret bounded by* $\tilde{O}(n^{3/4}\sqrt{dmCT})$.

---

**Algorithm 3** Randomized Rounding with Madow's Sampling Scheme

---

**Input:** Fractional cache allocation $(\boldsymbol{y}, \boldsymbol{z})$.
**Output:** A rounded allocation $(\hat{\boldsymbol{y}}, \hat{\boldsymbol{z}})$
 1: Let $\boldsymbol{y}$ be the output of the LP (9).
 2: **for** each cache $j \in \mathcal{J}$ **do**
 3:     Independently sample $S_j$ a set of $C$ files with the probability vector $\boldsymbol{y}^j$ using Madow's sampling scheme 4.
 4:     $\hat{y}_f^j \leftarrow \mathbb{1}(f \in S_j)$.
 5: **end for**
 6: $\hat{z}_f^i \leftarrow \bigvee_{j \in \partial^+(i)} \hat{y}_f^j, \forall i \in \mathcal{I}.$
 7: **Return** $(\hat{\boldsymbol{y}}, \hat{\boldsymbol{z}})$

---

## 5   Bounding the Number of Fetches

Fetching files from the remote server to the local caches consumes bandwidth and increases network congestion. Under non-prefetching policies, such as LRU, FIFO, and LFU, a file is fetched if and only if there is a cache miss. Hence, for these policies, it is enough to bound the cache miss rates in order to control the download rate. However, since the `LeadCache` policy decouples the fetching process from the cache misses, in addition to a small regret, we need to ensure that the number of file fetches remains small as well. We now prove the surprising result that if the file request process satisfies a mild regularity property, the file fetches *stop almost surely after a finite time*. Note that our result is of a different flavor from the long line of work that minimizes the switching regret under adversarial inputs but yield a much weaker bound on the number of fetches [Mukhopadhyay and Sinha, 2021, Devroye et al., 2015, Kalai and Vempala, 2005, Geulen et al., 2010].

**A. Stochastic Regularity Assumption:** Let $\{\boldsymbol{X}(t)\}_{t \geq 1}$ be the cumulative request-arrival process. We assume that, there exists a set of non-negative numbers $\{p_f^i\}_{i \in \mathcal{I}, f \in [N]}$ such that for any $\epsilon > 0$:

$$\sum_{t=1}^{\infty} \sum_{i \in \mathcal{I}, f \in [N]} \mathbb{P}\left(\left|\frac{\boldsymbol{X}_f^i(t)}{t} - p_f^i\right| \geq \epsilon\right) < \infty. \tag{19}$$

Using the first Borel-Cantelli Lemma, the regularity assumption **A** implies that the process $\{\boldsymbol{X}(t)\}_{t \geq 1}$ satisfies the strong-law: $\boldsymbol{X}_f^i(t)/t \to p_f^i$, a.s., $\forall i \in \mathcal{I}, f \in [N]$. However, the converse may not be true. Nevertheless, the assumption **A** is quite mild and holds, *e.g.*, when the file request sequence is generated by a renewal process having an inter-arrival distribution with a finite fourth moment (See Section 15 of the supplementary material for the proof). Define a fetch event $F(t)$ to take place at time slot $t$ if the cache configuration at time slot $t$ is different from that of at time slot $t - 1$, *i.e.*, $F(t) := \{\boldsymbol{y}(t) \neq \boldsymbol{y}(t-1)\}$. The following is our main result in this section:

**Theorem 4.** *Under the stochastic regularity assumption **A**, the file fetches to the caches stop after a finite time with probability 1 under the* LeadCache *policy.*

Please refer to Section 14 of the supplementary material for the proof of Theorem 4. This Theorem implicitly assumes that the optimization problems in steps 7 and 8 are solved exactly at every time.

# 6 A Minimax Lower Bound

In this section, we establish a minimax lower bound to the regret for the Bipartite Caching problem. Recall that our reward function (1) is non-linear. As such, the standard lower bounds, such as Theorem 5.1 and 5.12 of Orabona [2019], Theorem 5 of Abernethy et al. [2008] are insufficient for our purpose. The first regret lower bound for the Bipartite Caching problem was given by Bhattacharjee et al. [2020]. We now strengthen this bound by utilizing results from graph coloring.

**Theorem 5** (Regret Lower Bound). *For a catalog of size $N \geq \max(2\frac{d^2 C m}{n}, 2mC)$ the regret of any online caching policy $\pi$ is lower bounded as: $R_T^{\pi} \geq \max\left(\sqrt{\frac{mnCT}{2\pi}}, d\sqrt{\frac{mCT}{2\pi}}\right) - \Theta\left(\frac{1}{\sqrt{T}}\right)$.*

**Proof outline:** We use the standard probabilistic method where the worst-case regret is lower bounded by the average regret over an ensemble of problems. To lower-bound the cumulative reward accrued by the optimal offline policy, we construct an offline static caching configuration, where each user sees different files on each of the connected caches (*local exclusivity*). This construction effectively linearizes the reward function that can be analyzed using the probabilistic *Balls-into-Bins* framework and graph coloring theory. See Section 16 of the supplementary material for the proof.

**Approximation guarantee for** LeadCache**:** Theorem 1 and Theorem 5, taken together, imply that the LeadCache policy achieves the optimal regret within a factor of $\tilde{O}(\min((nd)^{1/4}, (n/d)^{3/4}))$. Hence, irrespective of $d$, the LeadCache policy is regret-optimal up to a factor of $\tilde{O}(n^{3/8})$.

# 7 Experiments

In this section, we compare the performance of the LeadCache policy with standard caching policies.

**Baseline policies:** Under the LRU policy [Borodin and El-Yaniv, 2005], each cache considers the set of all requested files from its connected users independently of other caches. In the case of a *cache-miss*, the LRU policy fetches the requested file into the cache while evicting a file that was requested *least recently*. The LFU policy works similarly to the LRU policy with the exception that, in the case of a cache-miss, the policy evicts a file that was requested the *least number of times* among all files currently on the cache. Finally, for the purpose of benchmarking, we also experiment with Belady's offline MIN algorithm [Aho et al., 1971], which is optimal in the class of non-pre-fetching reactive policies for each *individual* cache when the entire file request sequence is known a priori. Note that Belady's algorithm is *not* optimal in the network caching setting as it does not consider the adjacency relations between the caches and the users. The heuristic multi-cache policy by Bhattacharjee et al. [2020] uses an FTPL strategy for file prefetching while approximating the reward function (1) by a linear function.

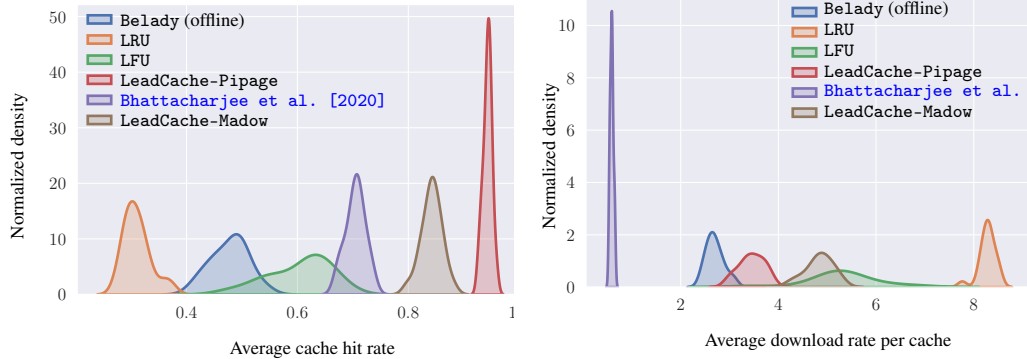

Figure 3: Empirical distributions of (a) Cache hit rates and (b) Fetch rates of different caching policies.

**Experimental Setup:** In our experiments, we use a publicly available anonymized production trace from a large CDN provider available under a BSD 2-Clause License [Berger et al., 2018, Berger, 2018]. The trace data consists of three fields, namely, request number, file-id, and file size. In our experiments, we construct a random Bipartite caching network with $n = 30$ users and $m = 10$ caches. Each cache is connected to $d = 8$ randomly chosen users. Thus, every user is connected to $\approx 2.67$ caches on average. The storage capacity $C$ of each cache is taken to be $10\%$ of the catalog size. We divide the trace consisting of the first $\sim 375K$ requests into 20 consecutive sub-intervals. File requests are assigned to the users sequentially before running the experiments on each of the sub-intervals. The code for the experiments is available online [Paria and Sinha, 2021].

**Results and Discussion:** Figure 3 shows the performance of different caching policies in terms of the average cache hit rate per file and the average number of file fetches per cache. From the plots, it is clear that the `LeadCache` policy, with any of its Pipage and Madow rounding variants, outperforms the baseline policies in terms of the cache hit rate. Furthermore, we find that the deterministic Pipage rounding variant empirically outperforms the randomized Madow rounding variant (that has a guaranteed sublinear $\alpha$-regret) in terms of both hit rate and fetch rates. On the flip side, the heuristic policy of Bhattacharjee et al. [2020] incurs a fewer number of file fetches compared to the `LeadCache` policy. From the plots, it is clear that the `LeadCache` policy excels by effectively coordinating the caching decisions among different caches and quickly adapting to the dynamic file request pattern. Section 17 of the supplementary material gives additional plots for the popularity distribution and the temporal dynamics of the policies for a given file request sequence.

## 8 Conclusion and Future Work

In this paper, we proposed an efficient network caching policy called `LeadCache`. We showed that the policy is competitive with the state-of-the-art caching policies, both theoretically and empirically. We proved a lower bound for the achievable regret and established that the number of file-fetches incurred by our policy is finite under reasonable assumptions. Note that `LeadCache` optimizes the cumulative cache hits without considering fairness among the users. In particular, `LeadCache` could potentially ignore a small subset of users who request unpopular content. A future research direction could be to incorporate fairness into the `LeadCache` policy so that *each* user incurs a small regret [Destounis et al., 2017]. Finally, it will be interesting to design caching policies that enjoy strong guarantees for the regret and the competitive ratio simultaneously [Daniely and Mansour, 2019].

## 9 Acknowledgement

This work was partially supported by the grant IND-417880 from Qualcomm, USA, and a research grant from the Govt. of India under the IoE initiative. The computational work reported on this paper was performed on the AQUA Cluster at the High Performance Computing Environment of IIT Madras. The authors would also like to thank Krishnakumar from IIT Madras for his help with a few illustrations appearing on the paper.

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
