## 10 Supplementary Material for the paper `LeadCache`: *Regret-Optimal Caching in Networks* by Debjit Paria and Abhishek Sinha

### 10.1 Proof of Lemma 1

From equation (4), we have that for any file request vector $\boldsymbol{x}$ and virtual caching configuration $\boldsymbol{z}$ :

$$r(\boldsymbol{x}, \boldsymbol{z}) = \langle \boldsymbol{x}, \boldsymbol{z} \rangle \leq q(\boldsymbol{x}, \psi(\boldsymbol{z})).$$

Thus for any file request sequence $\{\boldsymbol{x}_t\}_{t \geq 1}$, we have:

$$\sum_{t=1}^{T} r(\boldsymbol{x}_t, \boldsymbol{z}_t) \leq \sum_{t=1}^{T} q(\boldsymbol{x}_t, \psi(\boldsymbol{z}_t)). \tag{20}$$

On the other hand, let $\boldsymbol{y}_* \in \arg\max_{\boldsymbol{y} \in \mathcal{Y}} \sum_{t=1}^{T} q(\boldsymbol{x}_t, \boldsymbol{y})$ be an optimal static offline cache configuration vector corresponding to the file requests $\{\boldsymbol{x}_t\}_{t=1}^{T}$. Consider a candidate static virtual cache configuration vector $\boldsymbol{z}_* \in \mathcal{Z}$ defined as:

$$\boldsymbol{z}_*^i \equiv \min \left\{ \mathbf{1}_{N \times 1}, \Big( \sum_{j \in \partial^+(i)} \boldsymbol{y}_*^j \Big) \right\}, \quad 1 \leq i \leq n.$$

We have

$$\max_{\boldsymbol{z} \in \mathcal{Z}} \sum_{t=1}^{T} r(\boldsymbol{x}_t, \boldsymbol{z}) \geq \langle \sum_{t=1}^{T} \boldsymbol{x}_t, \boldsymbol{z}_* \rangle = \max_{\boldsymbol{y} \in \mathcal{Y}} \sum_{t=1}^{T} q(\boldsymbol{x}_t, \boldsymbol{y}). \tag{21}$$

Combining Eqns. (20) and (21), we conclude that

$$\max_{\boldsymbol{y} \in \mathcal{Y}} \sum_{t=1}^{T} q(\boldsymbol{x}_t, \boldsymbol{y}) - \sum_{t=1}^{T} q(\boldsymbol{x}_t, \psi(\boldsymbol{z}_t)) \leq \max_{\boldsymbol{z} \in \mathcal{Z}} \sum_{t=1}^{T} r(\boldsymbol{x}_t, \boldsymbol{z}) - \sum_{t=1}^{T} r(\boldsymbol{x}_t, \boldsymbol{z}_t). \tag{22}$$

Taking supremum of both sides of inequality (22) over all possible file request sequences $\{\boldsymbol{x}_t\}_{t \geq 1}$ yields the result. ∎

### 10.2 Proof of Theorem 1

Keeping Lemma 1 in view, to prove the desired regret upper bound for the `LeadCache` policy, it is enough to bound the regret for the virtual policy $\pi^{\text{virtual}}$ only. Following Cohen and Hazan [2015] we derive a general expression for the regret upper bound applicable to any linear reward function under an anytime `FTPL` policy. This is accomplished in the following steps. First, we extend the argument of Cohen and Hazan [2015] to the anytime setting. Then, we specialize this bound to our problem setting.

Recall the notations used in the paper - the aggregate file-request sequence from all users is denoted by $\{\boldsymbol{x}_t\}_{t \geq 1}$ and the virtual cache configuration sequence is denoted by $\{\boldsymbol{z}_t\}_{t \geq 1}$. Define the cumulative requests up to time $t$ as:

$$\boldsymbol{X}_t = \sum_{\tau=1}^{t-1} \boldsymbol{x}_\tau.$$

Note that the `LeadCache` policy chooses the virtual cache configuration at time slot $t$ by solving the following optimization problem at time slot $t$:

$$\boldsymbol{z}_t = \arg\max_{\boldsymbol{z} \in \mathcal{Z}} \langle \boldsymbol{z}, \boldsymbol{X}_t + \eta_t \boldsymbol{\gamma} \rangle, \tag{23}$$

where each of the $Nn$ components of the random vector $\boldsymbol{\gamma}$ is sampled independently from the standard Gaussian distribution, and $\mathcal{Z}$ denotes the set of all feasible virtual cache configurations as defined earlier in the paper. Next, we define the following potential function:

$$\Phi_{\eta_t}(\boldsymbol{x}) \equiv \mathbb{E}_{\boldsymbol{\gamma}} \left[ \max_{\boldsymbol{z} \in \mathcal{Z}} \langle \boldsymbol{z}, \boldsymbol{x} + \eta_t \boldsymbol{\gamma} \rangle \right]. \tag{24}$$

Since the perturbation r.v. $\boldsymbol{\gamma}$ is Gaussian, it follows that the potential function $\phi_{\eta_t}(\boldsymbol{x})$ is twice continuously differentiable [Abernethy et al., 2016, Lemma 1.5]. Furthermore, since the max function

is convex, we may interchange the expectation and gradient to obtain $\nabla \Phi_{\eta_t}(\boldsymbol{X}_t) = \mathbb{E}(\boldsymbol{z}_t)$ [Bertsekas, 1973, Proposition 2.2]. Thus we have:

$$\langle \nabla \Phi_{\eta_t}(\boldsymbol{X}_t), \boldsymbol{x}_t \rangle = \mathbb{E}\langle \boldsymbol{z}_t, \boldsymbol{x}_t \rangle. \tag{25}$$

To upper bound the regret of the `LeadCache` policy, we expand $\Phi_{\eta_t}(\boldsymbol{X}_{t+1})$ in second-order Taylor's series as follows:

$$
\begin{aligned}
&\Phi_{\eta_t}(\boldsymbol{X}_{t+1}) \\
=\ & \Phi_{\eta_t}(\boldsymbol{X}_t + \boldsymbol{x}_t) \\
=\ & \Phi_{\eta_t}(\boldsymbol{X}_t) + \langle \nabla \Phi_{\eta_t}(\boldsymbol{X}_t), \boldsymbol{x}_t \rangle + \frac{1}{2}\boldsymbol{x}_t^T \nabla^2 \Phi_{\eta_t}(\tilde{\boldsymbol{X}}_t)\boldsymbol{x}_t,
\end{aligned}
\tag{26}
$$

where $\tilde{\boldsymbol{X}}_t$ lies on the line segment joining $\boldsymbol{X}_t$ and $\boldsymbol{X}_{t+1}$. Plugging in the expression of the inner product from Eqn. (25) in expression (26), we obtain:

$$\mathbb{E}\langle \boldsymbol{z}_t, \boldsymbol{x}_t \rangle = \Phi_{\eta_t}(\boldsymbol{X}_{t+1}) - \Phi_{\eta_t}(\boldsymbol{X}_t) - \frac{1}{2}\boldsymbol{x}_t^T \nabla^2 \Phi_{\eta_t}(\tilde{\boldsymbol{X}}_t)\boldsymbol{x}_t. \tag{27}$$

Summing up Eqn. (27) from $t = 1$ to $T$, the total expected reward accrued by the `LeadCache` policy may be computed to be:

$$
\begin{aligned}
&\mathbb{E}\big(Q^{\text{LeadCache}}(T)\big) \\
=\ & \sum_{t=1}^{T} \mathbb{E}\langle \boldsymbol{z}_t, \boldsymbol{x}_t \rangle \\
=\ & \sum_{t=1}^{T}\bigg(\Phi_{\eta_t}(\boldsymbol{X}_{t+1}) - \Phi_{\eta_t}(\boldsymbol{X}_t)\bigg) - \frac{1}{2}\sum_{t=1}^{T}\boldsymbol{x}_t^T \nabla^2 \Phi_{\eta_t}(\tilde{\boldsymbol{X}}_t)\boldsymbol{x}_t \\
=\ & \sum_{t=1}^{T}\bigg(\Phi_{\eta_t}(\boldsymbol{X}_{t+1}) - \Phi_{\eta_{t+1}}(\boldsymbol{X}_{t+1}) + \Phi_{\eta_{t+1}}(\boldsymbol{X}_{t+1}) - \Phi_{\eta_t}(\boldsymbol{X}_t)\bigg) - \frac{1}{2}\sum_{t=1}^{T}\boldsymbol{x}_t^T \nabla^2 \Phi_{\eta_t}(\tilde{\boldsymbol{X}}_t)\boldsymbol{x}_t \\
=\ & \sum_{t=1}^{T}\bigg(\Phi_{\eta_t}(\boldsymbol{X}_{t+1}) - \Phi_{\eta_{t+1}}(\boldsymbol{X}_{t+1})\bigg) + \Phi_{\eta_{T+1}}(\boldsymbol{X}_{T+1}) - \Phi_{\eta_1}(\boldsymbol{X}_1) - \frac{1}{2}\sum_{t=1}^{T}\boldsymbol{x}_t^T \nabla^2 \Phi_{\eta_t}(\tilde{\boldsymbol{X}}_t)\boldsymbol{x}_t.
\end{aligned}
$$

Next, note that

$$
\begin{aligned}
\Phi_{\eta_{T+1}}(\boldsymbol{X}_{T+1}) \quad &= \quad \mathbb{E}_{\gamma}\big[\max_{\boldsymbol{z}\in\mathcal{Z}}\langle \boldsymbol{z}, \boldsymbol{X}_{T+1} + \eta_{T+1}\boldsymbol{\gamma}\rangle\big] \\
&\overset{\text{(Jensen's ineq.)}}{\geq} \quad \max_{\boldsymbol{z}\in\mathcal{Z}}\big[\mathbb{E}_{\gamma}\langle \boldsymbol{z}, \boldsymbol{X}_{T+1} + \eta_{T+1}\boldsymbol{\gamma}\rangle\big] \\
&= \quad \max_{\boldsymbol{z}\in\mathcal{Z}}\langle \boldsymbol{z}, \boldsymbol{X}_{T+1}\rangle \\
&= \quad Q^*(T),
\end{aligned}
$$

where recall that $Q^*(T)$ denotes the optimal cumulative reward up to time $T$ obtained by the best static policy in hindsight. Hence, from the above, we can upper bound the expected regret (2) of the `LeadCache` policy as:

$$
\begin{aligned}
&\mathbb{E}(R_T^{\text{LeadCache}}) \\
=\ & Q^*(T) - \mathbb{E}\big(Q^{\text{LeadCache}}(T)\big) \\
\leq\ & \Phi_{\eta_1}(\boldsymbol{X}_1) + \underbrace{\sum_{t=1}^{T}\bigg(\Phi_{\eta_{t+1}}(\boldsymbol{X}_{t+1}) - \Phi_{\eta_t}(\boldsymbol{X}_{t+1})\bigg)}_{(a)} + \frac{1}{2}\sum_{t=1}^{T}\boldsymbol{x}_t^T \nabla^2 \Phi_{\eta_t}(\tilde{\boldsymbol{X}}_t)\boldsymbol{x}_t.
\end{aligned}
\tag{28}
$$

**Bounding the term (a):** Next, to upper bound the expected regret, we control term (a) in inequality (28). From Eqns. (23) and (24), we can write:

$$\Phi_{\eta_{t+1}}(\boldsymbol{X}_{t+1}) = \mathbb{E}\big[\langle \boldsymbol{z}_{t+1}, \boldsymbol{X}_{t+1} + \eta_{t+1}\boldsymbol{\gamma}\rangle\big],$$

and

$$\Phi_{\eta_t}(\boldsymbol{X}_{t+1}) \geq \mathbb{E}\big[\langle \boldsymbol{z}_{t+1}, \boldsymbol{X}_{t+1} + \eta_t\boldsymbol{\gamma}\rangle\big].$$

Hence, each term in the summation (a) may be upper bounded as follows:

$$
\begin{aligned}
\Phi_{\eta_{t+1}}(\boldsymbol{X}_{t+1}) - \Phi_{\eta_t}(\boldsymbol{X}_{t+1}) &\leq \mathbb{E}\big[\langle \boldsymbol{z}_{t+1}, \boldsymbol{X}_{t+1} + \eta_{t+1}\boldsymbol{\gamma}\rangle\big] - \mathbb{E}\big[\langle \boldsymbol{z}_{t+1}, \boldsymbol{X}_{t+1} + \eta_t\boldsymbol{\gamma}\rangle\big] \\
&= \mathbb{E}\big[\langle \boldsymbol{z}_{t+1}, (\eta_{t+1} - \eta_t)\boldsymbol{\gamma})\rangle\big] \\
&= (\eta_{t+1} - \eta_t)\mathbb{E}\big[\langle \boldsymbol{z}_{t+1}, \boldsymbol{\gamma}\rangle\big] \\
&\leq (\eta_{t+1} - \eta_t)\mathbb{E}\big[\max_{z \in \mathcal{Z}}\langle \boldsymbol{z}, \boldsymbol{\gamma}\rangle\big] \\
&= (\eta_{t+1} - \eta_t)\mathcal{G}(\mathcal{Z}),
\end{aligned}
$$

where the quantity $\mathcal{G}(\mathcal{Z})$ is known as the *Gaussian complexity* of the set $\mathcal{Z}$ of virtual configurations Wainwright [2019]. Since the Gaussian perturbation $\boldsymbol{\gamma}$ has zero mean, $\mathcal{G}(\mathcal{Z})$ is non-negative due to Jensen's inequality. Substituting the above upper bound back in Eqn. (28), we notice that the summation in part (a) telescopes, yielding the following bound for the expected regret:

$$
\mathbb{E}(R_T^{\texttt{LeadCache}}) \leq \eta_{T+1}\underbrace{\mathcal{G}(\mathcal{Z})}_{(b)} + \frac{1}{2}\sum_{t=1}^{T}\underbrace{\boldsymbol{x}_t^T \nabla^2 \Phi_{\eta_t}(\tilde{\boldsymbol{X}}_t)\boldsymbol{x}_t}_{(c)}. \tag{29}
$$

We now upper bound each of the terms (b) and (c) as defined in the above regret bound.

**Bounding term (b) in Eqn. (29):** In the following, we upper bound the Gaussian complexity of the set $\mathcal{Z}$:

$$
\mathcal{G}(\mathcal{Z}) \equiv \mathbb{E}_{\boldsymbol{\gamma}}\big[\max_{\boldsymbol{z} \in \mathcal{Z}}\langle \boldsymbol{z}, \boldsymbol{\gamma}\rangle\big].
$$

From equation (4), we have for any feasible $\boldsymbol{z} \in \mathcal{Z}$:

$$
\begin{aligned}
\sum_{i \in \mathcal{I}, f \in [N]} \boldsymbol{z}_f^i &\leq \sum_{i \in \mathcal{I}, f \in [N]} \sum_{j \in \partial^+(i)} \boldsymbol{y}_f^j \\
&= \sum_{j \in \mathcal{J}} \sum_{f \in [N]} \sum_{i \in \partial^-(j)} \boldsymbol{y}_f^j \\
&\overset{(a)}{\leq} d \sum_{j \in \mathcal{J}} \sum_{f \in [N]} \boldsymbol{y}_f^j \\
&\overset{(b)}{\leq} dmC. \tag{30}
\end{aligned}
$$

where, in step (a), we have used our assumption that the right-degree of the bipartite graph $\mathcal{G}$ is upper bounded by $d$, and in (b), we have used the fact that the capacity of each cache is bounded by $C$.

For any fixed $\boldsymbol{z} \in \mathcal{Z}$, the random inner-product $\langle \boldsymbol{z}, \boldsymbol{\gamma}\rangle$ follows a normal distribution with mean zero and variance $\sigma^2$ where

$$
\sigma^2 \equiv \mathbb{E}\langle \boldsymbol{z}, \boldsymbol{\gamma}\rangle^2 \overset{(a)}{=} \sum_{i \in \mathcal{I}, f \in [N]} (\boldsymbol{z}_f^i)^2 \overset{(b)}{=} \sum_{i \in \mathcal{I}, f \in [N]} \boldsymbol{z}_f^i \overset{(c)}{\leq} dmC.
$$

In the above, equality (a) follows from the fact that $\boldsymbol{\gamma}$ is a standard normal r.v., equality (b) follows from the fact that the components $z_f^i$'s are binary-valued (hence, $(z_f^i)^2 = z_f^i$), and equality (c) follows from the upper bound given in Eqn. (30).

Next, observe that since the feasible set $\mathcal{Z}$ is downward closed, if $\boldsymbol{z}_* \in \arg\max_{\boldsymbol{z} \in \mathcal{Z}}\langle \boldsymbol{z}, \boldsymbol{\gamma}\rangle$, then $\gamma_f^i < 0$ implies $z_{*f}^i = 0$. Hence, we can simplify the expression for the Gaussian complexity of the set $\mathcal{Z}$ as

$$
\mathcal{G}(\mathcal{Z}) \equiv \mathbb{E}_{\boldsymbol{\gamma}}\big[\max_{\boldsymbol{z} \in \mathcal{Z}}\langle \boldsymbol{z}, \boldsymbol{\gamma}\rangle\big] = \mathbb{E}_{\boldsymbol{\gamma}}\big[\max_{\boldsymbol{z} \in \mathcal{Z}}\sum_{(i,f):\gamma_f^i > 0} z_f^i \gamma_f^i\big].
$$

Since all coefficients $\gamma_f^i$ in the above summation are positive, we conclude that there exists an optimal vector $\boldsymbol{z}_* \in \mathcal{Z}$ such that the inequality in Eqn. (4) is met with an equality for other components of $\boldsymbol{z}$, i.e., $\forall (i, f) : \gamma_f^i > 0$, we have

$$
\boldsymbol{z}_{*f}^i = \min(1, \sum_{j \in \partial^+(i)} y_{*f}^j), \tag{31}
$$

for some $\boldsymbol{y}_* \in \mathcal{Y}$. Let $\mathcal{Z}_*$ be the set of all feasible virtual caching vectors satisfying (31) for some feasible $\boldsymbol{y}_* \in \mathcal{Y}$. Since the optimal virtual caching vector $\boldsymbol{z}_* \in \mathcal{Z}_*$ is completely determined by the corresponding physical caching vector $\boldsymbol{y}_* \in \mathcal{Y}$, we have that $|\mathcal{Z}_*| \leq |\mathcal{Y}|$. Furthermore, since any of the $m$ caches can be loaded with any $C$ files, we have the bound:

$$|\mathcal{Y}| \leq \binom{N}{C}^m \leq \left(\frac{Ne}{C}\right)^{mC}, \tag{32}$$

where the last inequality is a standard upper bound for binomial coefficients. Finally, using Massart [2007]'s lemma for Gaussian variables, we have

$$\mathcal{G}(\mathcal{Z}) \equiv \mathbb{E}_{\boldsymbol{\gamma}}\big[\max_{\boldsymbol{z} \in \mathcal{Z}}\langle \boldsymbol{z}, \boldsymbol{\gamma}\rangle\big] = \mathbb{E}_{\boldsymbol{\gamma}}\big[\max_{\boldsymbol{z} \in \mathcal{Z}_*} \sum_{(i,f):\gamma_f^i > 0} z_f^i \gamma_f^i\big] \leq \sqrt{dmC}\sqrt{2\log|\mathcal{Z}_*|}$$

$$\leq mC\sqrt{2d\left(\log\frac{N}{C} + 1\right)}. \tag{33}$$

**Bounding term (c) in Eqn. (29):** Let us denote the file requested by the $i^{\text{th}}$ user at time $t$ by $f_i$. Using Abernethy et al. [2016, Lemma 1.5], we have

$$\big(\nabla^2 \Phi_{\eta_t}(\tilde{\boldsymbol{X}}_t)\big)_{\boldsymbol{pq}} = \frac{1}{\eta_t}\mathbb{E}\big[\hat{z}_{\boldsymbol{p}}\gamma_{\boldsymbol{q}}\big], \tag{34}$$

where $\hat{\boldsymbol{z}} \in \arg\max_{\boldsymbol{z} \in \mathcal{Z}}\langle \boldsymbol{z}, \tilde{\boldsymbol{X}}_t + \eta_t\boldsymbol{\gamma}\rangle$, and each of the indices $\boldsymbol{p}, \boldsymbol{q}$ is a (user, file) tuple. Hence, using Eqn. (34), and noting that each user requests only one file at a time, we have:

$$\begin{aligned}
\boldsymbol{x}_t^T \nabla^2 \Phi_{\eta_t}(\tilde{\boldsymbol{X}}_t)\boldsymbol{x}_t &= \frac{1}{\eta_t}\sum_{i,j \in \mathcal{I}}\mathbb{E}\big[\hat{z}_{f_i}^i \gamma_{f_j}^j\big] \\
&= \frac{1}{\eta_t}\mathbb{E}\big(\sum_{i \in \mathcal{I}}\hat{z}_{f_i}^i\big)\big(\sum_{j \in \mathcal{I}}\gamma_{f_j}^j\big) \\
&\overset{(a)}{\leq} \frac{1}{\eta_t}\sqrt{\mathbb{E}\big(\sum_{i \in \mathcal{I}}\hat{z}_{f_i}^i\big)^2 \mathbb{E}\big(\sum_{j \in \mathcal{I}}\gamma_{f_j}^j\big)^2} \\
&\overset{(b)}{\leq} \frac{1}{\eta_t}\sqrt{n^2 \times n} \\
&= \frac{1}{\eta_t}n^{3/2}, \tag{35}
\end{aligned}$$

where the inequality (a) follows from the Cauchy-Schwartz inequality and the inequality (b) follows from the facts that $\boldsymbol{z}$ are binary variables and that the components of the random vector $\boldsymbol{\gamma}$ are i.i.d. Finally, substituting the upper bounds from Eqns. (33) and (35) in the regret upper bound in Eqn. (29), we may upper bound the expected regret of the LeadCache policy as:

$$\begin{aligned}
\mathbb{E}\big(R_T^{\texttt{LeadCache}}\big) &\leq \eta_{T+1}\mathcal{G}(\mathcal{Z}) + \frac{n^{3/2}}{2}\sum_{t=1}^{T}\frac{1}{\eta_t} \\
&\leq \eta_{T+1}mC\sqrt{2d\left(\log\frac{N}{C} + 1\right)} + \frac{n^{3/2}}{2}\sum_{t=1}^{T}\frac{1}{\eta_t},
\end{aligned}$$

where the bound in the last inequality follows from Eqn. (33). Choosing the learning rates $\eta_t = \beta\sqrt{t}$ with an appropriate constant $\beta > 0$ yields the following regret upper bound:

$$\mathbb{E}\big(R_T^{\texttt{LeadCache}}\big) \leq kn^{3/4}d^{1/4}\sqrt{mCT},$$

for some $k = O(\text{poly-log}(N/C))$. ∎

## 11 Proof of Theorem 2

Denote the objective function of Problem (8) by $L(\boldsymbol{y}) \equiv \sum_{i,f}\theta_f^i \min(1, \sum_{j \in \partial^+(i)} y_f^j)$, where, to simplify the notations, we have not explicitly shown the dependence of the $\boldsymbol{\theta}$ coefficients on the time

index $t$. Recall the definition of surrogate objective function $\phi(\boldsymbol{y})$ given in Eqn. (15):

$$\phi(\boldsymbol{y}) = \sum_{i,f} (\theta_f^i)^+ \Big(1 - \prod_{j \in \partial^+(i)} (1 - y_f^j)\Big), \tag{36}$$

From Ageev and Sviridenko [2004, Eqn. (16)], we have the following algebraic inequality:

$$L(\boldsymbol{y}) \overset{(a)}{\geq} \phi(\boldsymbol{y}) \geq \Big(1 - (1 - \frac{1}{\Delta})^\Delta\Big) L(\boldsymbol{y}), \tag{37}$$

where $\Delta \equiv \max_{i \in \mathcal{I}} |\partial^+(i)|$. Note that inequality (a) holds with equality for binary vectors $\boldsymbol{y} \in \{0,1\}^{mN}$.

Let $\boldsymbol{y}^*$ be a solution of the relaxed LP (9), and OPT be the optimal value of the problem (8). Obviously, $L(\boldsymbol{y}^*) \geq$ OPT, which, combined with the estimate in Eqn. (37), yields:

$$\phi(\boldsymbol{y}^*) \geq \Big(1 - (1 - \frac{1}{\Delta})^\Delta\Big) \text{OPT}. \tag{38}$$

Since $\boldsymbol{y}^*$ is a solution to the relaxed LP, it may possibly contain fractional coordinates. In the following, we show that the Pipage rounding procedure, described in Algorithm 2, rounds at least one fractional variable of a cache at a round *without* decreasing the value of the surrogate objective function $\phi(\cdot)$ (Steps 3-6).

For a given fractional allocation vector $\boldsymbol{y}$, and another vector $\boldsymbol{v_y}$ of our choice depending on $\boldsymbol{y}$, define a one-dimensional function $g_{\boldsymbol{y}}(\cdot)$ as:

$$g_{\boldsymbol{y}}(s) = \phi(\boldsymbol{y} + s\boldsymbol{v_y}). \tag{39}$$

The vector $\boldsymbol{v_y}$ denotes the direction along which the fractional allocation vector $\boldsymbol{y}$ is rounded in the current step. The Pipage rounding procedure, Algorithm 2, chooses the vector $\boldsymbol{v_y}$ as follows: consider any cache $j$ that has at least two fractional coordinates $y_{f_1}^j$ and $y_{f_2}^j$ in the current allocation $\boldsymbol{y}$ (Step 3 of Algorithm 2) [2]. Take $\boldsymbol{v_y} = e_{j,f_1} - e_{j,f_2}$, where $e_{j,l}$ denotes the standard unit vector with 1 in the coordinate corresponding to the $l^{\text{th}}$ file of the $j^{\text{th}}$ cache, $l = f_1, f_2$. We now claim that the function $g_{\boldsymbol{y}}(s) = \phi(\boldsymbol{y} + s\boldsymbol{v_y})$ is linear in $s$. To see this, consider any one of the constituent terms of $g_{\boldsymbol{y}}(s)$ as given in Eqn. (36). Examining each term, we arrive at the following two cases:

1. If both $f \neq f_1$ and $f \neq f_2$ then that term is independent of $s$.

2. If either $f = f_1$, or $f = f_2$, the variables $y_{f_1}^j$ or $y_{f_2}^j$ may appear in each product term in (36) at most once. Since the product terms contain exactly one variable corresponding to each file, the variables $y_{f_1}^j$ and $y_{f_2}^j$ can not appear in the same product term together.

The above two cases imply that the function $g_{\boldsymbol{y}}(s)$ is linear in $s$. By increasing and decreasing the variable $s$ to the maximum extent possible, so that the candidate allocation $\boldsymbol{y} + s\boldsymbol{v_y}$ does not violate the constraint (12), we construct two new candidate allocation vectors $\boldsymbol{\alpha} = \boldsymbol{y} - \epsilon_1 \boldsymbol{v_y}$ and $\boldsymbol{\beta} = \boldsymbol{y} + \epsilon_2 \boldsymbol{v_y}$, where the constants $\epsilon_1$ and $\epsilon_2$ are chosen in such a way that at least one of the fractional variables of $\boldsymbol{y}$ becomes integral (Steps 4-5). It is easy to see that, by design, all cache capacity constraints in Eqn. (11) continue to hold in both of these two candidate allocations. In step 6, we choose the best of the candidate allocations $\boldsymbol{\alpha}$ and $\boldsymbol{\beta}$, corresponding to the surrogate function $\phi(\cdot)$. Let $\boldsymbol{y}^{\text{new}}$ denote the new candidate allocation vector. Since the maximum of a linear function over an interval is achieved on one of its two boundaries, we conclude that $\phi(\boldsymbol{y}^{\text{new}}) \geq \phi(\boldsymbol{y})$. As argued above, the rounded solution is feasible and has at least one less fractional coordinate. Hence, by repeated application of the above procedure, we finally arrive at a feasible integral allocation $\hat{\boldsymbol{y}}$ such that:

$$L(\hat{\boldsymbol{y}}) = \phi(\hat{\boldsymbol{y}}) \geq \phi(\boldsymbol{y}^*) \geq \Big(1 - (1 - \frac{1}{\Delta})^\Delta\Big) \text{OPT},$$

where the first equality follows from that fact that the functions $\phi(\boldsymbol{y}) = L(\boldsymbol{y})$ on integral points. ∎

---

[2] Since the cache capacities are integers, there cannot be a cache with only one fractional allocation variable.

## 12 Madow's Sampling Scheme

Madow's sampling scheme is a simple statistical procedure for randomly sampling a subset of items without replacement from a larger universe with a specified set of inclusion probabilities [Madow et al., 1949]. The pseudocode for Madow's sampling scheme is given in Algorithm 4. It samples $C$ items without replacement from a universe with $N$ items such that the item $i$ is included in the sampled set with probability $p_i, 1 \leq i \leq N$. The inclusion probabilities satisfy the feasibility constraint given by Eqn. (18).

---

**Algorithm 4** Madow's Systematic Sampling Scheme without Replacement

---

**Input:** A universe $[N]$ of size $N$, cardinality of the sampled set $C$, marginal inclusion probability vector $\boldsymbol{p} = (p_1, p_2, \ldots, p_N)$ satisfying the feasibility condition (18),
**Output:** A random set $S$ with $|S| = C$ such that, $\mathbb{P}(i \in S) = p_i, \forall i \in [N]$
 1: Define $\Pi_0 = 0$, and $\Pi_i = \Pi_{i-1} + p_i, \forall 1 \leq i \leq N$.
 2: Sample a uniformly distributed random variable $U$ from the interval $[0, 1]$.
 3: $S \leftarrow \phi$
 4: **for** $i \leftarrow 0$ to $C - 1$ **do**
 5:     Select element $j$ if $\Pi_{j-1} \leq U + i < \Pi_j$.
 6:     $S \leftarrow S \cup \{j\}$.
 7: **end for**
 8: **Return** $S$

---

**Correctness:** The correctness of Madow's sampling scheme is easy to verify. Due to the feasibility condition (18), Algorithm 4 selects exactly $C$ elements. Furthermore, the element $j$ is selected if the random variable $U$ falls in the interval $\sqcup_{i=1}^{N}[\Pi_{j-1} - i, \Pi_j - i)$. Since $U$ is uniformly distributed in $[0, 1]$, the probability that the element $j$ is selected is equal to $\Pi_j - \Pi_{j-1} = p_j, \forall j \in [N]$.

## 13 Proof of Proposition 1

The proof of the regret bound with the relaxed action set $\mathcal{Z}_{\text{rel}}$ follows the same line of arguments as the proof of Theorem 1 with integral cache allocations. In particular, we decompose the regret bound as in Eqn. (29), with the difference that we now replace the feasible set $\mathcal{Z}$ in term (b) with the relaxed feasible set $\mathcal{Z}_{\text{rel}}$. Observe that, for bounding the term (c), we did not exploit the fact that the variables are integral. Hence, the bound (35) holds in the case of the relaxed feasible set as well. However, for bounding the Gaussian complexity in term (b) in the proof of Theorem 1, we explicitly used the fact that the cache allocations (and hence, the virtual actions) are integral (*viz.* the counting argument in Eqn. (32)). To get around this issue, we now give a different argument for bounding the Gaussian complexity in term (b) for the relaxed action set $\mathcal{Z}_{\text{rel}}$. Note that for any feasible $(\boldsymbol{z}, \boldsymbol{y}) \in \mathcal{Z}_{\text{rel}}$, we have

$$\|\boldsymbol{z}\|_1 = \sum_{i,f} z_f^i \leq \sum_i \sum_{j \in \partial^+(i), f} y_f^j \leq d \sum_j \sum_f y_f^j \leq mCd, \tag{40}$$

where we have used the fact that each cache is connected to at most $d$ users and that each cache can hold $C$ files at a time. Hence, we have

$$\mathcal{G}(\mathcal{Z}_{\text{rel}}) = \mathbb{E}_{\boldsymbol{\gamma}}\big[ \max_{\boldsymbol{z} \in \mathcal{Z}_{\text{rel}}} \langle \boldsymbol{z}, \boldsymbol{\gamma} \rangle \big] \overset{\text{(Hölder's ineq.)}}{\leq} \mathbb{E}_{\boldsymbol{\gamma}}\big[ \max_{\boldsymbol{z} \in \mathcal{Z}_{\text{rel}}} \|\boldsymbol{z}\|_1 \|\boldsymbol{\gamma}\|_\infty \big] \leq mCd\sqrt{4\ln(Nn)}, \tag{41}$$

where, in the last inequality, we have used the $\ell_1$-norm bound (40) along with a standard upper bound on the expectation of the maximum of the absolute value of a set of i.i.d. standard Gaussian random variables [Wainwright, 2019]. Now proceeding similarly as in the proof of the regret bound for the action set $\mathcal{Z}$, we conclude that with an appropriate learning rate sequence, we have the following regret upper bound for the relaxed action set $\mathcal{Z}_{\text{rel}}$:

$$\mathbb{E}(\tilde{R}_T^{\text{LeadCache}}) \leq \kappa_1 n^{3/4}\sqrt{dmCT},$$

for some polylogarithmic factor $\kappa_1$. ∎

# 14 Proof of Theorem 4

**Discussion:** To intuitively understand why the total number of fetches is expected to be small under the `LeadCache` policy, consider the simplest case of a single cache with a single user [Bhattacharjee et al., 2020]. At every slot, the `LeadCache` policy populates the cache with a set of $C$ files having the highest perturbed cumulative count of requests $\boldsymbol{\Theta}(t)$. For the sake of argument, assume that the learning rate $\eta_t$ is time-invariant. Since at most one file is requested by the user per slot, only one component of $\boldsymbol{\Theta}(t)$ changes at a slot, and hence, the `LeadCache` policy fetches at most *one* new file at any time slot. Surprisingly, the following rigorous argument proves a far stronger result: the total number of fetches over an infinite time interval remains almost surely finite, even with a time-varying learning rate in the network caching setting.

**Proof:** Recall that, under the `LeadCache` policy, the optimal virtual caching configuration $\boldsymbol{z}_t$ for the $t^{\text{th}}$ slot is obtained by solving the optimization problem P:

$$\max_{\boldsymbol{z} \in \mathcal{Z}} \sum_{i \in \mathcal{I}} \langle \boldsymbol{\theta}^i(t), \boldsymbol{z}^i \rangle, \tag{42}$$

where we assume that the ties (if any) are broken according to some *fixed* tie-breaking rule. As discussed before, the corresponding physical cache configuration $\boldsymbol{y}_t$ may be obtained using the mapping $\psi(\cdot)$. Now consider a static virtual cache configuration $\tilde{\boldsymbol{z}}$ obtained by replacing the perturbed-count vectors $\boldsymbol{\theta}^i(t)$ in the objective function (42) with the vectors $\boldsymbol{p}^i, \forall i \in \mathcal{I}$, where $\boldsymbol{p} = (\boldsymbol{p}^i, i \in \mathcal{I})$ is defined to be the vector of long-term file-request probabilities, given by Eqn. (19). In other words,

$$\tilde{\boldsymbol{z}} \in \arg\max_{\boldsymbol{z} \in \mathcal{Z}} \sum_{i \in \mathcal{I}} \langle \boldsymbol{p}^i, \boldsymbol{z}^i \rangle. \tag{43}$$

Since the set of all possible virtual caching configurations $\mathcal{Z}$ is finite, the objective value corresponding to any other non-optimal caching configuration must be some non-zero gap $\delta > 0$ away from that of an optimal configuration. Let us denote the set of all *sub-optimal* virtual cache configuration vectors by $\mathcal{B}$. Hence, for any $\boldsymbol{z} \in \mathcal{B}$, we must have:

$$\sum_{i \in \mathcal{I}} \langle \boldsymbol{p}^i, \tilde{\boldsymbol{z}}^i \rangle \geq \sum_{i \in \mathcal{I}} \langle \boldsymbol{p}^i, \boldsymbol{z}^i \rangle + \delta. \tag{44}$$

Let us define an "error" event $\mathcal{E}(t)$ to be event such that the `LeadCache` policy yields a sub-optimal virtual cache configuration (and possibly, a sub-optimal physical cache configuration (4)) at time $t$. Let $G$ be a zero-mean Gaussian random variable with variance $2Nn$. We now upper bound the probability of the error event $\mathcal{E}(t)$ as below:

$$\mathbb{P}(\mathcal{E}(t))$$

$$\overset{(a)}{\leq} \mathbb{P}\left( \sum_{i \in \mathcal{I}} \langle \boldsymbol{\theta}^i(t), \boldsymbol{z}^i(t) \rangle > \sum_{i \in \mathcal{I}} \langle \boldsymbol{\theta}^i(t), \tilde{\boldsymbol{z}}^i \rangle, \boldsymbol{z}(t) \in \mathcal{B} \right)$$

$$\overset{(b)}{\leq} \mathbb{P}\left( \eta_t G \geq \sum_{i \in \mathcal{I}} \langle \boldsymbol{X}^i(t), \tilde{\boldsymbol{z}}^i \rangle - \sum_{i \in \mathcal{I}} \langle \boldsymbol{X}^i(t), \boldsymbol{z}^i(t) \rangle, \boldsymbol{z}(t) \in \mathcal{B} \right)$$

$$\overset{(c)}{=} \mathbb{P}\left( \eta_t G \geq \sum_{i \in \mathcal{I}} \langle \boldsymbol{X}^i(t), \tilde{\boldsymbol{z}}^i \rangle - \sum_{i \in \mathcal{I}} \langle \boldsymbol{X}^i(t), \boldsymbol{z}^i(t) \rangle, \sum_{i \in \mathcal{I}} \langle \boldsymbol{X}^i(t), \tilde{\boldsymbol{z}}^i \rangle - \sum_{i \in \mathcal{I}} \langle \boldsymbol{X}^i(t), \boldsymbol{z}^i(t) \rangle > \frac{\delta t}{2}, \boldsymbol{z}(t) \in \mathcal{B} \right)$$

$$+ \mathbb{P}\left( \eta_t G \geq \sum_{i \in \mathcal{I}} \langle \boldsymbol{X}^i(t), \tilde{\boldsymbol{z}}^i \rangle - \sum_{i \in \mathcal{I}} \langle \boldsymbol{X}^i(t), \boldsymbol{z}^i(t) \rangle, \sum_{i \in \mathcal{I}} \langle \boldsymbol{X}^i(t), \tilde{\boldsymbol{z}}^i \rangle - \sum_{i \in \mathcal{I}} \langle \boldsymbol{X}^i(t), \boldsymbol{z}^i(t) \rangle \leq \frac{\delta t}{2}, \boldsymbol{z}(t) \in \mathcal{B} \right)$$

$$\overset{(d)}{\leq} \mathbb{P}\left( \eta_t G \geq \frac{\delta t}{2} \right) + \mathbb{P}\left( \frac{1}{t} \sum_{i \in \mathcal{I}} \langle \boldsymbol{X}^i(t), \tilde{\boldsymbol{z}}^i - \boldsymbol{z}^i(t) \rangle \leq \frac{\delta}{2}, \boldsymbol{z}(t) \in \mathcal{B} \right)$$

$$= \mathbb{P}\left( \eta_t G \geq \frac{\delta t}{2} \right) + \mathbb{P}\left( \sum_{i \in \mathcal{I}} \langle \frac{\boldsymbol{X}^i(t)}{t} - \boldsymbol{p}^i, \tilde{\boldsymbol{z}}^i - \boldsymbol{z}^i(t) \rangle + \sum_{i \in \mathcal{I}} \langle \boldsymbol{p}^i, \tilde{\boldsymbol{z}}^i - \boldsymbol{z}^i(t) \rangle \leq \frac{\delta}{2}, \boldsymbol{z}(t) \in \mathcal{B} \right)$$

$$\overset{(e)}{\leq} \mathbb{P}\left( \eta_t G \geq \frac{\delta t}{2} \right) + \mathbb{P}\left( \sum_{i \in \mathcal{I}} \langle \frac{\boldsymbol{X}^i(t)}{t} - \boldsymbol{p}^i, \tilde{\boldsymbol{z}}^i - \boldsymbol{z}^i(t) \rangle \leq -\frac{\delta}{2} \right)$$

$$\overset{(f)}{\leq} \mathbb{P}\left( \eta_t G \geq \frac{\delta t}{2} \right) + \mathbb{P}\left( \sum_{i \in \mathcal{I}} \sum_{f \in [N]} \left| \frac{\boldsymbol{X}_f^i(t)}{t} - p_f^i \right| \geq \frac{\delta}{2} \right)$$

$$\overset{(g)}{\leq} \quad \mathbb{P}\left(\eta_t G \geq \frac{\delta t}{2}\right) + \mathbb{P}\left(\bigcup_{i,f} \left|\frac{\boldsymbol{X}_f^i(t)}{t} - p_f^i\right| \geq \frac{\delta}{2Nn}\right)$$

$$\overset{(h)}{\leq} \quad \mathbb{P}\left(\eta_t G \geq \frac{\delta t}{2}\right) + \sum_{i,f} \mathbb{P}\left(\left|\frac{\boldsymbol{X}_f^i(t)}{t} - p_f^i\right| \geq \frac{\delta}{2Nn}\right)$$

$$\overset{(i)}{\leq} \quad \exp(-ct) + Nn\alpha_\epsilon(t).$$

for some positive constants $c$ and $\epsilon$, which depend on the problem parameters.

In the above chain of inequalities:

(a) follows from the fact that on the error event $\mathcal{E}(t)$, the virtual cache configuration vector $\boldsymbol{z}(t)$ must be in the sub-optimal set $\mathcal{B}$ and, by definition, it must yield more objective value in (42) than the optimal virtual cache configuration vector $\tilde{\boldsymbol{z}}$,
(b) follows by writing $\boldsymbol{\Theta}(t) = \boldsymbol{X}(t) + \eta_t\boldsymbol{\gamma}$, and observing that the virtual configurations $\boldsymbol{z}(t) \in \mathcal{B}$ and $\tilde{\boldsymbol{z}}(t)$ may differ in at most $Nn$ coordinates, and that the normal random variables are increasing (in the convex ordering sense) with their variances,
(c) follows from the law of total probability,
(d) follows from the monotonicity of the probability measures,
(e) follows from Eqn. (44),
(f) follows from the fact that for any two equal-length vectors $\boldsymbol{a}, \boldsymbol{b}$, triangle inequality yields:

$$\langle \boldsymbol{a}, \boldsymbol{b}\rangle \geq -\sum_k |\boldsymbol{a}_k||\boldsymbol{b}_k|,$$

and that $|\tilde{\boldsymbol{z}}_f^i - \boldsymbol{z}_f^i(t)| \leq 1, \forall i, f$,
(g) follows from the simple observation that at least one number in a set of some numbers must be at least as large as the average,
(h) follows from the union bound,
and finally, the inequality (i) follows from the concentration inequality for Gaussian variables and the concentration inequality for the request process $\{\boldsymbol{X}(t)\}_{t\geq 1}$, as given by Eqn. (19). Using the above bound and the assumptions on the request sequence, we have

$$\sum_{t\geq 1} \mathbb{P}(\mathcal{E}(t)) \leq \sum_{t\geq 1} \exp(-ct) + Nn \sum_{t\geq 1} \alpha_\epsilon(t) < \infty.$$

Hence, the first Borel-Cantelli Lemma implies that

$$\mathbb{P}(\mathcal{E}(t) \text{ i.o}) = 0.$$

Hence, almost surely, the error events stop after a finite time. Thus, with a fixed tie-breaking rule, the new file fetches stop after a finite time w.p. $1$. ∎

## 15 Renewal Processes satisfies the Regularity Condition (A)

Suppose that, for any $i \in \mathcal{I}, f \in [N]$, the cumulative request process $\{\boldsymbol{X}_f^i(t)\}_{t\geq 1}$ constitutes a renewal process such that the renewal intervals have a common expectation $1/p_f^i$ and a finite fourth moment. Let $S_k$ be the time of the $k^{\text{th}}$ renewal, $k \geq 1$ [Ross, 1996]. In other words, the $i^{\text{th}}$ user requests file $f$ for the $k^{\text{th}}$ time at time $S_k, k \geq 1$. Then we have

$$\mathbb{P}\left(\frac{\boldsymbol{X}_f^i(t)}{t} - p_f^i \leq -\epsilon\right) \quad = \quad \mathbb{P}\left(\boldsymbol{X}_f^i(t) \leq t(p_f^i - \epsilon)\right)$$

$$\leq \quad \mathbb{P}\left(S_{\lfloor t(p_f^i - \epsilon)\rfloor} \geq t\right)$$

$$\leq \quad \mathbb{P}\left((S_{\lfloor t(p_f^i-\epsilon)\rfloor} - \lfloor t(p_f^i - \epsilon)\rfloor)^4 \geq (t(1 - p_f^i + \epsilon))^4\right)$$

$$\overset{(a)}{\leq} \quad O\left(\frac{1}{t^2}\right),$$

where, in (a), we have used the Markov Inequality along with a standard upper bound on the fourth moment of a centered random walk. Using a similar line of arguments, we can show that

$$\mathbb{P}\left(\frac{\boldsymbol{X}_f^i(t)}{t} - p_f^i \ge \epsilon\right) \le O(\frac{1}{t^2}).$$

Combining the above two bounds, we conclude that

$$\sum_{t \ge 1} \mathbb{P}\left(|\frac{\boldsymbol{X}_f^i(t)}{t} - p_f^i| \ge \epsilon\right) < \infty.$$

The above derivation verifies the regularity condition A for renewal request processes. ∎

## 16   Proof of Theorem 5

We establish a slightly stronger result by proving the announced lower bound for a regular bipartite network with uniform left-degree $d_L$ and uniform right-degree $d$. Counting the total number of edges in two different ways, we have $nd_L = md$. Hence, $d_L = \frac{md}{n}$. For pedagogical reasons, we divide the entire proof into several logically connected parts.

**(A) Some Observations and Preliminary Lemmas:**   To facilitate the analysis, we introduce the following surrogate linear reward function:

$$q_{\text{linear}}(\boldsymbol{x}, \boldsymbol{y}) \equiv \sum_{i \in \mathcal{I}} \boldsymbol{x}_t^i \cdot \Big( \sum_{j \in \partial^+(i)} \boldsymbol{y}_t^j \Big). \tag{45}$$

We begin our analysis with the following two observations:

1. **Upper Bound:** From the definition (1) of the rewards, we clearly have:

$$q(\boldsymbol{x}, \boldsymbol{y}) \le q_{\text{linear}}(\boldsymbol{x}, \boldsymbol{y}), \quad \forall \boldsymbol{x}, \boldsymbol{y}. \tag{46}$$

2. **Local Exclusivity implies Linearity:** In the case when all caches connected to each user host *different* files, *i.e.,* the cached files are *locally exclusive* in the sense that they are not duplicated from each user's local point-of-view, *i.e.,*

$$\boldsymbol{y}^{j_1} \cdot \boldsymbol{y}^{j_2} = 0, \quad \forall j_1 \ne j_2 : j_1, j_2 \in \partial^+(i), \forall i \in \mathcal{I}, \tag{47}$$

the reward function (1) reduces to a linear one:

$$q(\boldsymbol{x}, \boldsymbol{y}) = q_{\text{linear}}(\boldsymbol{x}, \boldsymbol{y}), \quad \forall \boldsymbol{x}, \boldsymbol{y}. \tag{48}$$

The equation (48) follows from the fact that with the local exclusivity constraint, we have

$$\sum_{j \in \partial^+(i)} \boldsymbol{y}_t^j \le \mathbf{1}, \quad \forall i \in \mathcal{I},$$

where the inequality holds component wise. Hence, the 'min' operator in the definition of the reward function (Eqn. (1)) is vacuous in this case. To make use of the linearity of the rewards as in Eqn. (48), the caches need to store items in such a way that the local exclusivity condition (47) holds. Towards this, we now define a special coloring of the nodes in the set $\mathcal{J}$ for a given bipartite graph $\mathcal{G}(\mathcal{I} \uplus \mathcal{J}, E)$.

**Definition 2** (Valid $\chi$-coloring of the caches)**.** *Let $\chi$ be a positive integer. A valid $\chi$-coloring of the caches of a bipartite network $\mathcal{G}(\mathcal{I} \uplus \mathcal{J}, E)$ is an assignment of colors from the set $\{1, 2, \ldots, \chi\}$ to the vertices in $\mathcal{J}$ (i.e., the caches) in such a way that all neighboring caches $\partial^+(i)$ to every node $i \in \mathcal{I}$ (i.e., the users) are assigned different colors.*

Obviously, for a given bipartite graph $\mathcal{G}$, a valid $\chi$-coloring of the caches exists only if the number of possible colors $\chi$ is large enough. The following lemma gives an upper bound to the value of $\chi$ so that a valid $\chi$-coloring of the caches exists.

**Lemma 2.** *Consider a bipartite network $\mathcal{G}(\mathcal{I} \cup \mathcal{J}, E)$, where each user $i \in \mathcal{I}$ is connected to at most $d_L$ caches, and each cache $j \in \mathcal{J}$ is connected to at most $d$ users. Then there exists a valid $\chi$-coloring of the caches where $\chi \leq d_L d$.*

*Proof.* From the given bipartite network $\mathcal{G}(V, E)$, construct another graph $H(V', E')$, where the caches form the vertices of $H$, *i.e.*, $V' \equiv \mathcal{J}$. For any two vertices in $u, v \in V'$, there is an edge $(u, v) \in E'$ if and only if a user $i \in \mathcal{I}$ is connected to both the caches $u$ and $v$ in the bipartite network $\mathcal{G}$. Next, consider any cache $j \in \mathcal{J}$. By our assumption, it is connected to at most $d$ users. On the other hand, each of the users is connected to at most $d_L - 1$ caches other than $j$. Hence, the degree of any node $j$ in the graph $H$ is upper bounded as:

$$\Delta' \leq d(d_L - 1) \leq d_L d - 1.$$

Finally, using Brook's theorem Diestel [2005], we conclude that the vertices of the graph $H$ may be colored using at most $1 + \Delta' \leq d_L d$ different colors. $\qquad \square$

**(B) Probabilistic Method for Regret Lower Bounds:** With the above results at our disposal, we now employ the well-known probabilistic method for proving the regret lower bound [Alon and Spencer, 2004]. The basic principle of the probabilistic method is quite simple. We compute a lower bound to the *expected regret* for any online network caching policy $\pi$ for a chosen joint probability distribution $p(\boldsymbol{x}_1, \boldsymbol{x}_2, \ldots, \boldsymbol{x}_T)$ over an *ensemble* of incoming file request sequence. Since the maximum of a set of numbers is at least as large as the expectation, the above quantity also gives a lower bound to the regret for the worst-case file request sequence. Clearly, the tightness of the resulting bound largely depends on our ability to identify a suitable input distribution $p(\cdot)$ that is amenable to analysis and, at the same time, yields a good bound. In the following, we show how this program can be elegantly carried out for the network caching problem.

Fix a valid $\chi$-coloring of the caches, and let $k = \chi C$. Consider a library consisting of $N = 2k$ different files. We now choose a randomized file request sequence $\{\boldsymbol{x}_t^i \equiv \boldsymbol{\alpha}_t\}_{t=1}^T$ where each user $i$ requests the same (random) file $\boldsymbol{\alpha}_t$ at slot $t$ such that the common file request vector $\boldsymbol{\alpha}_t$ is sampled uniformly at random from the set of the *first* $2k$ unit vectors $\{\boldsymbol{e}_i \in \mathbb{R}^{2k}, 1 \leq i \leq 2k\}$ independently at each slot [3]. Formally, we choose:

$$p(\boldsymbol{x}_1, \boldsymbol{x}_2, \ldots, \boldsymbol{x}_T) := \prod_{t=1}^T \left( \frac{1}{2k} \mathbb{1}\big(\boldsymbol{x}_t^{i_1} = \boldsymbol{x}_t^{i_2}, \ \forall i_1, i_2 \in \mathcal{I}\big) \right).$$

**(C) Upper-bounding the Total Reward accrued by any Online Policy:** Making use of observation (46), the expected total reward $G_T^\pi$ accrued by any online network caching policy $\pi$ may be upper bounded as follows:

$$
\begin{aligned}
G_T^\pi \quad &\leq \quad \mathbb{E}\left( \sum_{t=1}^T \sum_{i \in \mathcal{I}} \boldsymbol{x}_t^i \cdot \sum_{j \in \partial^+(i)} \boldsymbol{y}_t^j \right) \\
&\overset{(a)}{=} \quad \sum_{t=1}^T \sum_{(i,j) \in E} \mathbb{E}\big(\boldsymbol{x}_t^i \cdot \boldsymbol{y}_t^j\big) \\
&\overset{(b)}{=} \quad \sum_{t=1}^T \sum_{(i,j) \in E} \mathbb{E}\big(\boldsymbol{x}_t^i\big) \cdot \mathbb{E}\big(\boldsymbol{y}_t^j\big) \\
&\overset{(c)}{=} \quad \frac{1}{2k} \sum_{t=1}^T \mathbb{E}\left( \sum_{(i,j) \in E} \sum_{f \in [N]} \boldsymbol{y}_{tf}^j \right) \\
&\overset{(d)}{\leq} \quad \frac{d}{2k} \sum_{t=1}^T \mathbb{E}\left( \sum_{j \in \mathcal{J}} \sum_{f \in [N]} \boldsymbol{y}_{tf}^j \right) \\
&\overset{(e)}{\leq} \quad \frac{dmCT}{2k},
\end{aligned}
\tag{49}
$$

---

[3]Recall that, according to our one-hot encoding convention, a request for a file $f$ by any user corresponds to the unit vector $\boldsymbol{e}_f$.

where the eqn. (a) follows from the linearity of expectation, eqn. (b) follows from the fact that the cache configuration vector $\boldsymbol{y}_t$ is independent of the file request vector $\boldsymbol{x}_t$ at the same slot, as the policy is online, the eqn. (c) follows from the fact that each of the $N = 2k$ components of the vector $\mathbb{E}(\boldsymbol{x}_t^i)$ is equal to $\frac{1}{2k}$, the inequality (d) follows from the fact that each cache is connected to at most $d$ users, and finally, the inequality (e) follows from the cache capacity constraints.

**(D) Lower-bounding the Total Reward Accrued by the Static Oracle:** We now lower bound the total expected reward accrued by the optimal static offline policy (*i.e.,* the first term in the regret definition (2)). Note that, due to the presence of the 'min' operator in (1), obtaining an exactly optimal static cache configuration vector $\boldsymbol{y}^*$ is non-trivial, as it requires solving an **NP-hard** optimization problem (8) (with the vector $\boldsymbol{\theta}(t)$ in the objective replaced by the cumulative file request vector $\boldsymbol{X}(T) \equiv \sum_{t=1}^{T} \boldsymbol{x}_t$). However, since we only require a good lower bound to the total expected reward, a suitably constructed sub-optimal caching configuration will serve our purpose, provided that we can compute a lower bound to its expected reward. Towards this end, in the following, we construct a joint cache configuration vector $\boldsymbol{y}_\perp$ that satisfies the local exclusivity constraint (47).

**D.1 Construction of a "good" cache configuration vector $\boldsymbol{y}_\perp$:** Let $\mathcal{X}$ be a valid $\chi$-coloring of the caches. Let the color $c$ appear in $f_c$ different caches in the coloring $\mathcal{X}$. To simplify the notations, we relabel the colors in non-increasing order of their frequency of appearance in the coloring $\mathcal{X}$, *i.e.,*

$$f_1 \geq f_2 \geq \ldots \geq f_\chi. \tag{50}$$

Let the vector $\boldsymbol{v}$ be obtained by sorting the components of the vector $\sum_{t=1}^{T} \boldsymbol{\alpha}_t$ in non-increasing order. Partition the vector $\boldsymbol{v}$ into $\frac{2k}{C} = 2\chi$ segments by sequentially merging $C$ contiguous coordinates of $\boldsymbol{v}$ at a time. Let $c_j$ denote the color of the cache $j \in \mathcal{J}$ in the coloring $\mathcal{X}$. The cache configuration vector $\boldsymbol{y}_\perp$ is constructed by loading each cache $j \in \mathcal{J}$ with the set of all $C$ files in the $c_j^{\text{th}}$ segment of the vector $\boldsymbol{v}$. See Figure 4 for an illustration.

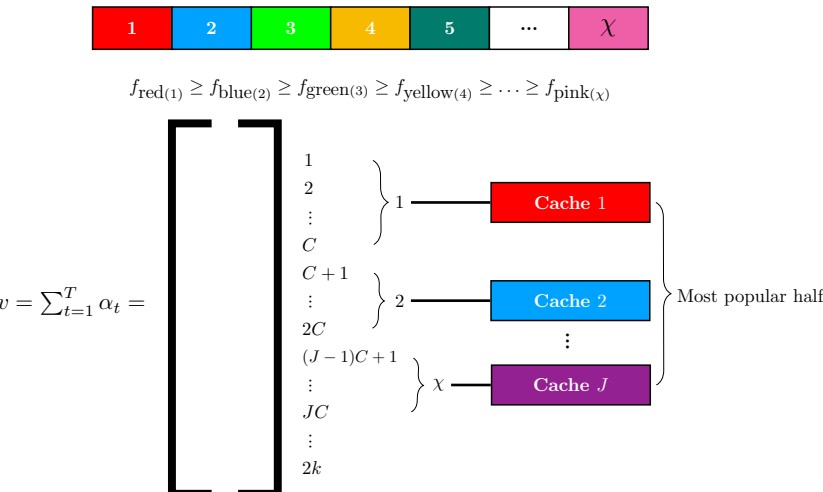

Figure 4: Construction of the caching configuration $\boldsymbol{y}_\perp$.

**D.2 Observation:** Since the vector $\boldsymbol{v}$ has $2\chi$ segments (each containing $C$ different files) and the number of possible colors in the coloring $\mathcal{X}$ is $\chi$, it follows that only the most popular half of the files get mapped to some caches under $\boldsymbol{y}_\perp$. Moreover, it can be easily verified that the cache configuration vector $\boldsymbol{y}_\perp$ satisfies the local exclusivity property (47).

**D.3 Analysis:** Let $S_{\boldsymbol{v}}(m)$ denote the sum of the frequency counts in the $m^{\text{th}}$ segment of the vector $\boldsymbol{v}$. In other words, $S_{\boldsymbol{v}}(m)$ gives the aggregate frequency of requests of the files in the $m^{\text{th}}$ segment of the vector $\boldsymbol{v}$. By construction, we have

$$S_{\boldsymbol{v}}(1) \geq S_{\boldsymbol{v}}(2) \geq \ldots \geq S_{\boldsymbol{v}}(\chi). \tag{51}$$

Since under the distribution $p$, all users request the same file at each time slot (*i.e.*, $x_t^i = \alpha_t$, $\forall i$), and since the linearity in rewards holds due to the local exclusivity property of the cache configuration $y_\perp$, the reward obtained by the files in the $j^{\text{th}}$ cache under the caching configuration $y_\perp$ is given by:

$$
\begin{aligned}
y_\perp^j \cdot \left( \sum_{i \in \partial^-(j)} \sum_{t=1}^T x_t^i \right) &= y_\perp^j \cdot \left( \sum_{i \in \partial^-(j)} \sum_{t=1}^T \alpha_t \right) \\
&\overset{(a)}{=} d y_\perp^j \cdot \left( \sum_{t=1}^T \alpha_t \right) \\
&\overset{(b)}{=} d S_v(c_j),
\end{aligned}
\tag{52}
$$

where the equation (a) follows from the fact that, in this converse proof, we are investigating a regular bipartite network where each cache is connected to exactly $d$ users, and the equation (b) follows from the construction of the cache configuration vector $y_\perp$. Hence, the expected aggregate reward accrued by the optimal stationary configuration $y^*$ may be lower-bounded by that of the configuration $y_\perp$ as follows:

$$
\begin{aligned}
G_T^{\pi^*} &\overset{(a)}{\geq} \mathbb{E}\left( \sum_{j \in \mathcal{J}} y_\perp^j \cdot \left( \sum_{i \in \partial^-(j)} \sum_{t=1}^T x_t^i \right) \right) \\
&\overset{(b)}{=} d \mathbb{E}\left( \sum_{j \in \mathcal{J}} S_v(c_j) \right) \\
&\overset{(c)}{=} d \mathbb{E}\left( \sum_{c=1}^{\mathcal{X}} f_c S_v(c) \right) \\
&\overset{(d)}{\geq} \frac{d}{\mathcal{X}} \left( \sum_{c=1}^{\mathcal{X}} f_c \right) \mathbb{E}\left( \sum_{c=1}^{\mathcal{X}} S_v(c) \right),
\end{aligned}
$$

where
(a) follows from the local exclusivity property of the configuration $y_\perp$,
(b) follows from Eqn. (52),
(c) follows after noting that the color $c$ appears on $f_c$ different caches in the coloring $\mathcal{X}$,
(d) follows from an algebraic inequality presented in Lemma 3 below, used in conjunction with the conditions (50) and (51).

Next, we lower bound the quantity $\mathbb{E}\left( \sum_{c=1}^{\mathcal{X}} S_v(c) \right)$ appearing on the right hand side of the equation (53). Conceptually, identify the catalog with $N = 2k$ "bins", and the random file requests $\{\alpha_t\}_{t=1}^T$ as "balls" thrown uniformly into one of the "bins." With this correspondence in mind, a little thought reveals that the random variable $\sum_{c=1}^{\mathcal{X}} S_v(c)$ is distributed identically as the total load in the most popular $k = \chi C$ bins when $T$ balls are thrown uniformly at random into $2k$ bins. Continuing with the above chain of inequalities, we have

$$
\begin{aligned}
G_T^{\pi^*} &\overset{(e)}{\geq} \frac{dmC}{k} \mathbb{E}\left( \text{load in the most popular half of } 2k \text{ bins with } T \text{ balls thrown u.a.r.} \right) \\
&\overset{(f)}{\geq} \frac{dmC}{k}\left( \frac{T}{2} + \sqrt{\frac{kT}{2\pi}} \right) - \Theta\left( \frac{1}{\sqrt{T}} \right) \\
&= \frac{dmCT}{2k} + dmC\sqrt{\frac{T}{2\pi k}} - \Theta\left( \frac{1}{\sqrt{T}} \right),
\end{aligned}
\tag{53}
$$

where, in the inequality (e), we have used the fact that $\sum_{c=1}^{\mathcal{X}} f_c = m$, and the inequality (f) follows from Lemma 4 stated below. Hence, combining Eqns. (49) and (53), and noting that $k = \chi C \leq d_L dC = \frac{mCd^2}{n}$ from Lemma 2, we conclude that for any caching policy $\pi$:

$$
R_T^\pi \geq G_T^{\pi^*} - G_T^\pi \geq \sqrt{\frac{mnCT}{2\pi}} - \Theta\left( \frac{1}{\sqrt{T}} \right).
\tag{54}
$$

Moreover, making use of a *Globally* exclusive configuration (where all caches store different files), in Theorem 7 of their paper, Bhattacharjee et al. [2020] proved the following regret lower bound for any

online caching policy $\pi$:

$$R_T^\pi \geq d\sqrt{\frac{mCT}{2\pi}} - \Theta\left(\frac{1}{\sqrt{T}}\right). \tag{55}$$

Hence, combining the bounds from Eqns. (54) and (55), we conclude that

$$R_T^\pi \geq \max\left(\sqrt{\frac{mnCT}{2\pi}}, d\sqrt{\frac{mCT}{2\pi}}\right) - \Theta\left(\frac{1}{\sqrt{T}}\right).$$

∎

---

**Lemma 3.** *Let $f_1 \geq f_2 \geq \ldots \geq f_n$ and $s_1 \geq s_2 \geq \ldots \geq s_n$ be two non-increasing sequences of $n$ real numbers each. Then*

$$\sum_{i=1}^n f_i s_i \geq \frac{1}{n}\left(\sum_{i=1}^n f_i\right)\left(\sum_{i=1}^n s_i\right).$$

---

*Proof.* From the rearrangement inequality (Hardy et al. [1952]), we have for each $0 \leq j \leq n-1$:

$$\sum_{i=1}^n f_i s_i \geq \sum_{i=1}^n s_i f_{(i+j)(\mathrm{mod}\, n)+1}, \tag{56}$$

where the modulo operator is used to cyclically shift the indices. Summing over the inequalities (56) for all $0 \leq j \leq n-1$, we have

$$n \sum_{i=1}^n f_i s_i \geq \left(\sum_{i=1}^n f_i\right)\left(\sum_{i=1}^n s_i\right),$$

which yields the result. □

---

**Lemma 4** (Bhattacharjee et al. [2020]). *Suppose that $T$ balls are thrown independently and uniformly at random into $2C$ bins. Let the random variable $M_C(T)$ denote the number of balls in the most populated $C$ bins. Then*

$$\mathbb{E}(M_C(T)) \geq \frac{T}{2} + \sqrt{\frac{CT}{2\pi}} - \Theta\left(\frac{1}{\sqrt{T}}\right).$$

---

## 17 Additional Experimental Results

In this section, we compare the performance of the `LeadCache` policy (with Pipage rounding) with other standard caching policies on two datasets taken from two different application domains. We observe that the relative performance of the algorithms remains qualitatively the same across the datasets, with the `LeadCache` policy consistently maintaining the highest hit rate. In our experiments, we instantiated a randomly generated bipartite network with $n = 30$ users and $m = 10$ caches. Each cache is connected to $d = 8$ randomly chosen users. The capacity of each cache is taken to be $10\%$ of the library size. Our experiments are run on HPE Apollo XL170rGen10 Servers with Dual Intel Xeon Gold 6248 20-core and 192 GB RAM.

### 17.1 Experiments with the CMU dataset [Berger et al., 2018]

**Dataset description:** This dataset is obtained from the production trace of a commercial content distribution network. It consists of 500M requests for a total of 18M objects. The popularity distribution of the requests follows approximately a Zipf distribution with the parameter $\alpha$ between 0.85 and 1. Since we are interested only in the order in which the requests arrive, we ignore the size

Table 1: Performance Evaluation with the CMU dataset [Berger et al., 2018]

| Policies | Hit Rate | Fetch Rate |
|---|---|---|
| LeadCache (with Pipage rounding) | **0.864** | **1.754** |
| LRU | 0.472 | 13.375 |
| LFU | 0.504 | 13.643 |
| Belady (offline) | 0.581 | 5.128 |

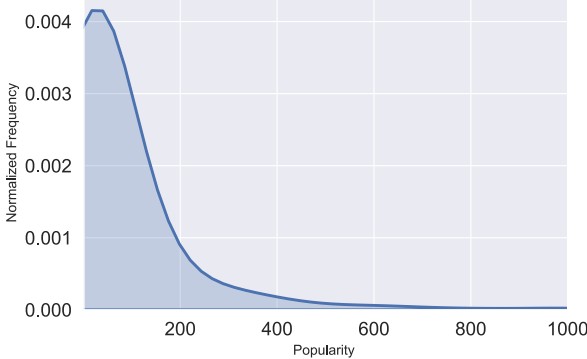

Figure 5: Plot showing the popularity distribution of the files in the CMU dataset

of the objects in our experiments. Due to the massive volume of the original dataset, we consider only the first ~ 375K requests in our experiments.

Figure 5 shows the sorted overall popularity distribution of the most popular files in the dataset. It is easy to see that the popularity distribution has a light tail - a small fraction of the files are extremely popular, while others are rarely requested. The *Recall distance* measures the number of file requests between two successive requests of the same file. Figure 8 shows a plot of the empirical Recall distance distribution for this dataset.

**Experimental Results:** Figure 6 compares the dynamics of the caching policies for a particular file request pattern. It shows that the proposed LeadCache policy maintains a high cache hit rate right from the beginning. In other words, the proposed policy quickly learns the file request pattern from all users and distributes the files near-optimally on different caches. This plot also shows that the fetch rate of the LeadCache policy remains small compared to the other three caching policies. Figure 7 gives a bivariate plot of the cache hits and downloads by the LeadCache policy. From the plots, it is clear that the LeadCache policy outperforms the benchmarks on this dataset in terms of

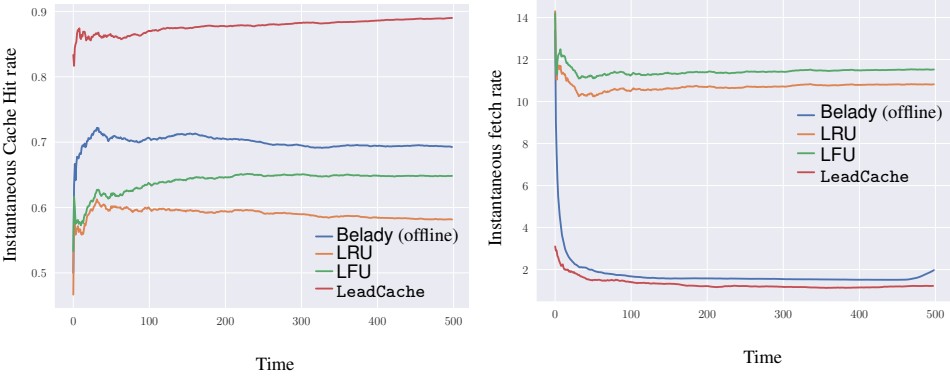

Figure 6: Temporal dynamics of instantaneous (a) cache hit rates and (b) fetch rates of different caching policies for a given file request sequence taken from the CMU dataset [Berger et al., 2018]

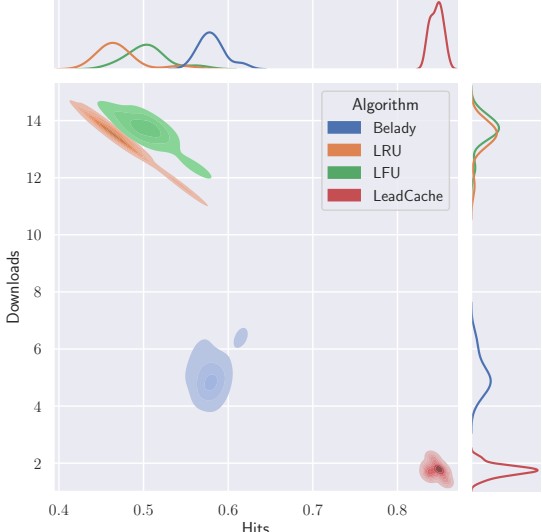

Figure 7: Bivariate plot of cache hit rates and the Fetch rates of different caching policies for the CMU dataset [Berger et al., 2018]

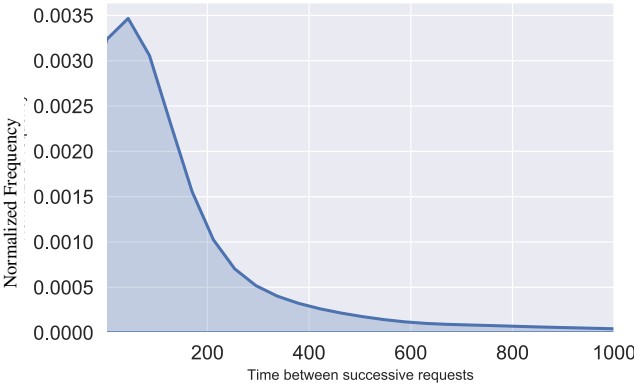

Figure 8: Distribution of time between two successive request of the same file on the CMU Dataset

both Hit rate and Fetch rate. The average values of the performance indices are summarized in Table 1.

## 17.2 Experiments with the MovieLens Dataset [Harper and Konstan, 2015]

**Dataset Description:** MovieLens [4] is a popular dataset containing ~ 20M ratings for $N$ ~ 27278 movies along with the timestamps of the ratings [Harper and Konstan, 2015]. The ratings were assigned by 138493 users over a period of approximately twenty years. Our working assumption is that a user rates a movie in the same sequence as she requests the movie file for download from the Content Distribution Network. Due to the sheer size of the dataset, in our experiments, we consider the first 1M ratings only. Figure 9 shows the empirical distribution of the number of times the movies have been rated (and hence, downloaded) by the users. Figure 10 shows the empirical distribution of time between two successive requests of the same file (*i.e.,* the *Recall distance*).

---

[4]This dataset is freely available from 

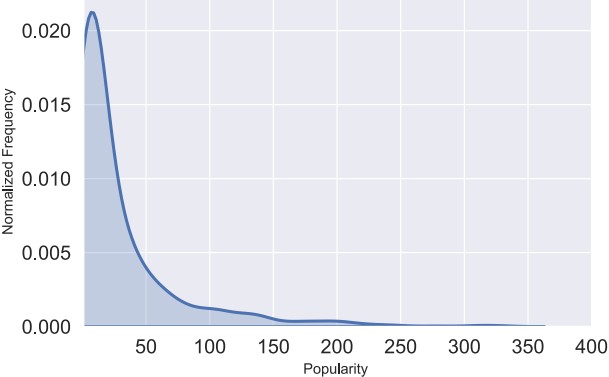

Figure 9: Empirical Popularity distribution of the number of ratings for the MovieLens Dataset [Harper and Konstan, 2015]

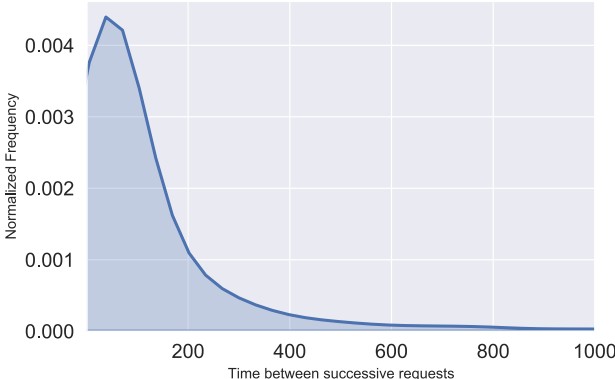

Figure 10: Distribution of time between two successive request of the same file on the MovieLens Dataset

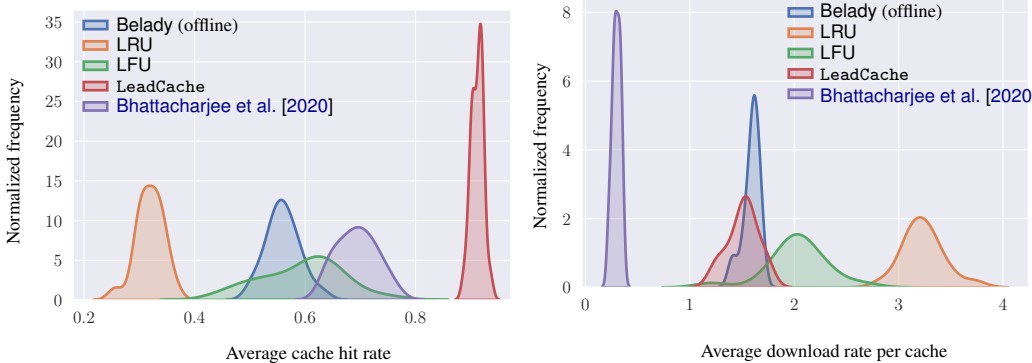

Figure 11: Empirical distributions of (a) Cache hit rates and (b) Fetch rates of different caching policies on the MovieLens Dataset.

Table 2: Performance Evaluation with the MovieLens dataset [Harper and Konstan, 2015]

| Policies | Hit Rate | Fetch Rate |
|---|---|---|
| LeadCache (with Pipage rounding) | **0.991** | 1.509 |
| Heuristic [Bhattacharjee et al., 2020] | 0.694 | **0.297** |
| LRU | 0.312 | 3.234 |
| LFU | 0.595 | 2.028 |
| Belady (offline) | 0.560 | 1.589 |

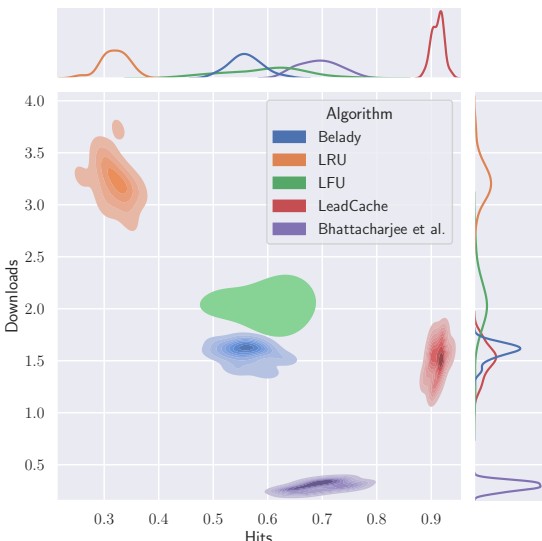

Figure 12: Bivariate plot of cache hit rates and the fetch rates of different caching policies for the MovieLens dataset [Harper and Konstan, 2015]

**Experimental Results**    Figure 11 compares the performance of different policies in terms of the hit rates and fetch rates. The average values of the key performance indicators are shown in Table 2. From the plots and the table, we see that the LeadCache policy achieves the highest hit rate among all other policies, which is about 32% more than that of the Heuristic policy proposed by Bhattacharjee et al. [2020]. On the other hand, it incurs more file fetches compared to only the heuristic policy proposed by Bhattacharjee et al. [2020]. Figure 12 gives a joint plot of the hit rate and the fetch rate of different policies. It is clear from the plots that the LeadCache policy robustly learns the file request patterns and caches them on the caches near-optimally.