# OpenReview forum: "$\texttt{LeadCache}$: Regret-Optimal Caching in Networks"
_NeurIPS.cc/2021/Conference — NeurIPS 2021 Poster_

### Official Review · Reviewer_txkX · 2021-07-14

**Rating:** 6
**Confidence:** 3

**Summary:**

This paper develops online algorithms for a bipartite caching problem. The problem is considered in an adversarial setting and the regret of the proposed algorithm is analyzed. Since the underlying problem is combinatorial, an intermediate step is needed to facilitate the algorithm design and regret analysis. This is done by designing another approximation algorithm with approximation analysis. Also, the regret lower bound of the problem is obtained, which improved the existing regret lower bound that we proposed recently in e Bhattacharjee et al. [2020]. Last, the performance is empirically evaluated using multiple datasets and against multiple traditional and recent regret-based algorithms.

**Limitations And Societal Impact:**

yes

**Main Review:**

Overall, this paper is well-written and well-organized. The technical presentation is clear and easy to follow. The results sound solid and the techniques that are used are different from the most related paper (Bhattacharjee et al. [2020]). It seems that the major technical contributions of the paper are the fast approximation algorithm and the improved regret lower bound. While the problem has been recently getting attention, it seems that the contribution of this paper is clear and substantial in the literature.

The only concern is how much the results are useful beyond the current problem. It seems that the approximation algorithm is highly utilizing the structure of the problem, in item 2 of Section 3, the authors claim that this is a new technique that could be useful beyond the current setting. But, in the rest of the paper, there are limited insights on why this is the case.

Minor comments:

-- It is better to briefly introduce the k-set problem in Line 27.

-- The cache capacity is fixed and equal to $C$ for all caches. Is it easy to relax this assumption?

-- The third contribution in Lines 148-153 should be states more clearly. The sublinearity in regret typically means that the decision converges to the best static action, leading to zero switching cost. However, the authors in this part claim that with some additional assumptions this happens. In addition, the regret result of the proposed algorithm is missing in Section 3.

-- Line 28, it seems that the policy $\pi$ is different from the $\pi$ notation in the lower bound equation.

-- It is true that the optimal offline for the networked caching problem is open, but, it is relatively straightforward to calculate this value when the problem size is small. This is missing in experiments and could help to show how much gap is still there between the proposed algorithm and the offline optimum.

-- Last, it is always good to plot the regret of the algorithm since it shows another aspect of the algorithms on how it evolves over time as compared to the theoretical analysis.

**Time Spent Reviewing:**

4

---

> ### Author Response · Authors · 2021-08-09
> **Thanks for a detailed review and constructive comments**
>
> **1. On the applicability of the techniques beyond the current setting:** We thank the reviewer for a sound summary of the work presented in the paper. While we agree that the particular Pipage rounding technique described in the paper is specific to the network caching problem, we argue that the general transformation technique of reducing a non-linear reward function to a linear one is useful beyond this particular problem. In the following, we give another application of this transformation technique to illustrate its efficacy.
>
>
> **Problem statement:** Consider a caching setting where a single user is connected to a single cache of capacity $C$. However, as opposed to the single-cache problem studied in [1], now assume that the user is interested in simultaneously retrieving **two** distinct files of its choice at a round (say an audio file and a background video file). Thus, the user receives a unit reward if **both** the requested files are stored in the cache or otherwise receives zero rewards. As before, our objective is to design an online caching policy with minimal regret.
>
> **Online Policy:** Let us denote the user's request at the time slot $t$ by an $N \times N$ dimensional symmetric matrix $X^t$, where $X^t_{ij}=X^t_{ji}=1$ if the pair $(i,j)$ was requested by the user at time $t$ and $X^t_{ij}=0$ otherwise. Denoting the set of files cached by the online policy at time $t$ by $y_t$, we can express the one-slot reward function as:
>
> \begin{eqnarray*}
> r_t = \frac{1}{2}\sum_{i, j} X^t_{ij} y_iy_j.
> \end{eqnarray*}
>
> Due to the non-linearity of the reward function, the FTPL policy cannot be applied to this problem directly. To get around this difficulty, we can analogously define an $N\times N$-dimensional symmetric **virtual action** matrix $Z$, such that $Z_{ij} = y_i y_j, \forall i,j$. With this transformation, the slot-wise reward vector takes the standard bilinear form $r_t = Trace(X^t \cdot z_t),$ to which FTPL can be applied. Let us denote the set of all feasible virtual actions as $\mathcal{Z} = \{ Z: Z_{ij} = y_i y_j, \sum_{i} y_i = C, y_i \in \{0,1\} \}.$ The FTPL policy needs to solve a linear optimization problem over the virtual action set $\mathcal{Z}.$ Similar to the problem studied in the paper, it is not difficult to show that this optimization problem is actually **NP-Hard** (Reduction from the Densest $k$-subgraph problem [2]). Let $\Theta_t$ be the cumulative perturbed request count vector (viz. step 6 of Algorithm 1) and let $\mathcal{S}^N_+$ denote the cone of all $N\times N$ real symmetric positive semidefinite matrices.  Mathematically, the FTPL problem can be formulated as follows.
>
> \begin{eqnarray*}
>  Maximize~ Trace(\Theta_t \cdot Z)
> \end{eqnarray*}
> Subject to,
> \begin{eqnarray*}
> Z &\in& S^N_+,\\\\
> rank(Z) &=& 1,\\\\
> Z_{ii} &\in& \{0,1\}, \forall i \\\\
> Trace(Z) &\leq& C.
> \end{eqnarray*}
> The constraints can be explained as follows: a rank-$1$ positive semidefinite matrix $Z$ can be decomposed as $yy^T$ for some real vector $y.$ The third constraint forces the vector $y$ to be binary. Finally, the fourth constraint follows from the cache capacity constraint as shown below:
> \begin{eqnarray*}
> Trace(Z)= \sum_{i}y_i^2 = \sum_i y_i \leq C,
> \end{eqnarray*}
> where we have used the fact that for a binary variable $y \in \{0,1\}$, we have $y^2=y.$
>
> Similar to the paper, one can suitably relax the non-convex constraints of the above program and attempt to design an $\alpha$-regret policy using low-rank matrix recovery techniques. Please see our response to the reviewer ZWvx in this connection. We will add a summary of the above example in the revised version.
>
> **2. Statement of the $k$-sets problem:** Assume that there is a pool of $N$ items. The $k$-set problem asks to design an online policy that selects a sequence of subsets consisting of $k$ items each so that a sequence of items requested by an adversary are included in the corresponding subsets the maximum number of times [2]. We will add this definition to the revised version.
>
>  **3. On the fixed cache capacity:**
> Yes, the online policy remains exactly the same (with the right-hand side of the constraint inequality (11) replaced by $C_j$). We assumed all caches to have the same capacity only to save on the notations. We will clarify this point in the revised version.
>
> **4. On the sub-linearity of regret and convergence of actions:** We respectfully beg to differ. The sub-linearity of the regret of an online policy means that the **cumulative reward** of the policy approaches to that of the best-fixed action in hindsight. However, for an arbitrary request sequence, an online policy can potentially keep switching its decisions forever while, at the same time, achieving a sublinear regret. As a simple counterexample, consider the classical prediction with experts advice problem [3] with two experts. Assume that the losses for the experts alternate between 0 and 1 in a complementary fashion, i.e., the loss sequence for the first expert is $010101...$ and that of the second expert is $101010...$. In this case, although the best-fixed action in hindsight is to stick to any one of the experts, many regret-optimal policies, e.g., Hedge, choose the experts roughly uniformly at random with probabilities 0.5 and 0.5 at each round. Hence the action of Hedge on this loss sequence never converges, conceding a linear switching cost.
>
> What we claim in our third contribution is that, with the stochastic request generation process, the above situation cannot occur with the LeadCache policy almost surely. Thus, the actions of the LeadCache policy converge almost surely under the stochastic assumption A on the input sequence.
>
> Since the regret is typically defined with respect to the worst-case sequence (without any probabilistic qualification, viz. Theorem 1), we did not mention another regret bound with the stochastic assumption. However, it is an immediate consequence of Theorem 3, that after a sufficiently long time, the regret of the LeadCache policy becomes **exactly zero** with probability one under the stochastic assumption A. We will state this fact after stating Theorem 3 in the revised version. Thanks for the pointer!
>
> **5. On notations:** In this paper, the symbol $\pi$ refers to any generic policy. Eqn. (2) defines the regret of any policy, and the lower bound in Theorem 4 is valid for any online policy. So, in general, they could be different.
>
> **6. On the gap between LeadCache and the offline optimum:** Thanks for the interesting suggestion. Although finding an exactly optimal offline configuration is still out of reach even for the small network that we simulated (it requires evaluating roughly $ \sim 10^{500}$ candidate solutions), it is not difficult to get an upper bound to the performance of the optimum static offline policy. One such bound is given directly by the LP (9) where no noise is added to the file frequency counts (i.e., in step 6 of Algorithm 1, we set $\eta_t=0$), and we use the cache-configuration returned by the LP at the last step, i.e., after receiving the file requests for the entire time horizon. The resulting solution yields an upper bound to the optimal offline reward because the LP may return fractional cache allocations. We will report this result in the revised version.
>
> **7. On the plot of the regret dynamics:** Thanks for the suggestion. Due to space limitations, we included a plot of the dynamics of the policy (in terms of the hit rate and fetch rates) in Figure 6 of the supplementary section. In the revised version, we will also include the corresponding regret plots.
>
>  **References:**
>
>  [1]. Bhattacharjee, Rajarshi, Subhankar Banerjee, and Abhishek Sinha. "Fundamental limits on the regret of online network-caching." Proceedings of the ACM on Measurement and Analysis of Computing Systems 4, no. 2 (2020): 1-31.
>
>  [2]. Cohen, Alon, and Tamir Hazan. "Following the perturbed leader for online structured learning." In International Conference on Machine Learning, pp. 1034-1042. PMLR, 2015.
>
>  [3] Cesa-Bianchi, Nicolo, and Gábor Lugosi. Prediction, learning, and games. Cambridge university press, 2006.

---

### Official Review · Reviewer_ocxm · 2021-07-15

**Rating:** 7
**Confidence:** 3

**Summary:**

The paper studies the Bipartite Caching problem, from an online learning perspective. A bipartite graph describes how n users are connected to m caches, each of which supports C files. At each time t, N files are distributed across m caches (same file can be placed in multiple caches), and each user requests one file. Every cache-hit (a user that requests a file in one of its caches) counts as a +1 towards the total reward. The goal is to maximize the reward (or minimize the regret, which is the difference with respect to the reward of an optimal policy).

The file demands are assumed to be chosen by an "oblivious" adversary, which means that the file demands for the entire time horizon T are chosen by an adversary, but a priori (i.e., before seeing the cache allocations).

The paper proposes a cache allocation algorithm called LeadCache. The algorithm is based on the "follow the perturbed leader" policy. In essence, at time t, we aggregate all previous user-file requests until time t-1, add Gaussian noise to them, and then perform the file allocation to caches as if the resulting perturbed vector was the current file demand. The paper builds on previous works that study the problem of Online Linear Optimization and provide regret bounds for "follow the perturbed leader" algorithms. However, in order to apply these ideas in the bipartite caching context, several new ideas need to be introduced. In particular, the file allocation for time t is found by first solving a linear relaxation of the combinatorial optimization problem, and then using an interesting rounding algorithm, for which a performance guarantee can be proven.

The paper provides a careful bound on the regret achieved by LeadCache and shows that it is at most a O(n^(3/8)) factor away from a minimax lower bound. Furthermore, the paper analyzes the number of file fetches needed by LeadCache (i.e., how many times a cache needs to "download" a new file) and shows that, interestingly, the file fetches stop after a finite amount of time.



**Limitations And Societal Impact:**

This line of research mainly seeks to improve caching systems. The only negative societal impact could be that the proposed optimal policy seeks to maximize the global reward, potentially at the expense of individual users who happen to request unpopular content. I asked above that the authors discuss potential drawbacks of this problem formulation.

**Main Review:**

The paper studies the interesting problem of caching, which is important from the point of view of cloud computing and content distribution. This problem has received attention from many angles (coding, queueing theory, etc.) and the online learning setting with an adversarial sequence of file demands is an interesting one.

While the paper builds on existing works on "follow the perturbed leader" policy, several new techniques need to be introduced to use this policy in the bipartite caching setting. In particular, the LP relaxation and the careful rounding to integers are pretty interesting. Furthermore, the regret analysis is fairly nontrivial and leads to a nice near-optimality guarantee. Finally, the result on the file fetching stopping after a while is interesting and the stochastic assumption used to establish it seems reasonable.

Overall, the paper is well written and clear, and I enjoyed reading it. It provides proof sketches and discusses some of the key technical ideas, and presents complete proofs in the appendix. While I didn't go through the proofs in the appendix carefully, I skimmed through them for the main ideas, and found the appendix to also be well written and clear.

The paper also provides a simulation analysis that compares LeadCache with other cache allocation policies. It shows that LeadCache outperforms classical policies in terms of cache-hit rate and file-fetches per cache. While these results seem to suggest that LeadCache is much better than the state-of-the-art, I think these results may be a bit misleading as these two metrics may not paint the whole picture. I mention more on that below.

My main concerns with the paper and the significance of the results have to do with the general formulation of the problem. From the point of view of caching for content distribution, it's a bit strange to me that the reward is just a function of the number of cache hits. In practice, if a user has a file demand and it is not met at time t, I would think that the system would have to eventually meet that demand (and all demands would have to be met within some delay). Under the setting in the paper, some users that happen to be in a bad spot in the network (sharing a cache with many other users) or that are requesting unpopular files may never have their demands met, since it may be optimal to "starve" some users in order to get cache-hits from other users. Therefore, I think that the regret metric only paints a part of the picture here, and the paper should discuss this.

Related to the point above, the fact that file-fetches stop after a while is presented as a positive point, but it seems to be a consequence of the objective of minimizing the cache-hit regret. A caching policy that stops changing after a while (once it finds the optimal way to maximize cache-hits) doesn't necessarily feel very practical. I would like to see some of these points discussed in the paper.

Some other comments:

- The proof of Theorem 3 relies on using Borel-Cantelli to show that a certain event E cannot happen infinitely often. The event E, as I understand it, is the event that the policy chosen by LeadCache is suboptimal with respect to the asymptotic file demand probabilities p^i. This implies that after some finite time t, LeadCache must keep picking the optimal policy for (p^i) forever. I'm a bit confused by this. Isn't the problem of finding the optimal cache allocation for a set of demand probabilities (p^i) an NP-hard combinatorial optimization problem? I may be missing something here, but the authors may want to clarify this.

- Algorithm 2, the pipage rounding, seems to assume that while y is not integral, there must be a cache with at least two non-integral variables. I believe this is a consequence of the fact that the sum of the y_{f,j}'s for some cache j must equal C for any optimal solution to the LP. Otherwise, I guess it could be that only one entry is non-integral? I may have missed this, but I didn't see it mentioned.

- I think the fetches weren't defined very formally. While I believe I understand the concept, it may be good to define what a fetch means and when it happens.

Minor issues/typos:
- line 88: the x_t in the summation should be x_{\tau}
- line 122: should the (N choose k)^m be (N choose C)^m?
- line 139: should "problem P" be "problem $\psi$"?

----

**After rebuttal:**
The authors addressed my concerns well, and my positive impression of the paper is maintained.




**Time Spent Reviewing:**

6

---

> ### Author Response · Authors · 2021-08-09
> **Thanks for a detailed review and constructive suggestions**
>
> **1. On incorporating "fairness" into the problem formulation:** We agree with the Reviewer that maximizing the cumulative cache hits of all users is an important but only the first step towards designing a more practical and robust network caching policy. As the Reviewer rightly pointed out, a potential future research direction could be to incorporate a notion of fairness into the reward function. In this connection, we would like to mention that an analogous problem has been extensively studied in the scheduling literature under the topic of ``fair-scheduling.” See, e.g., the papers [1], [2], [3]. In this problem, in addition to maximizing the throughput (which is analogous to our objective of maximizing the cumulative cache-hits), one is also interested in allocating the available system resources (e.g., bandwidth) to the users in a “fair” manner so that all users receive at least a minimal level of service. The concept of fairness has been recently investigated in the context of coded caching as well [4]. Several indices of fairness exist (e.g., MinMax fairness), and many scheduling algorithms are known (e.g., Proportional Fair [5]), which are fair in the sense that they achieve some of these fairness indices.
>
> Taking inspiration from this body of work, one possible way to incorporate the notion of fairness into our caching framework would be to add a regularization term corresponding to one of the above fairness indices to the cumulative cache-hit metric. Our virtual action framework, which can handle non-linearities in the reward function, would be particularly suitable in designing an online policy for the resulting online problem. We will add a summary of the above discussion in the paper.
>
> **2. On the stopping of the file-fetches after a finite time:** Note that, in Theorem 3, the file fetches stop after a finite time only under the stochastic assumption (A) made on the file request process. This result does not hold under the general adversarial model, in which case the cache configuration keeps changing forever, tracking the changing adversarial requests. In the single cache setting, it has been recently shown [6] that the number of file fetches grows as $\Theta(\sqrt{T})$ under the adversarial model. We will discuss this point in the revised version of the paper.
>
> **3. On the proof of Theorem 3:** The Reviewer is correct in his observation. Note that the description of the LeadCache policy in Algorithm 1 assumes an **oracle access** to the optimization problem given in step 7. Theorem 1 and Theorem 3 refer to this exact version of the LeadCache policy. If the assumption of the oracle access to the optimization solver is relaxed, we can show a sub-linear $(1-1/e)$-regret guarantee with a randomized rounding algorithm. For more details on this, please see our response to the comment of Reviewer ZWvx. It might be possible to prove a version of Theorem 3 where an approximate solver is used in Step 7 of the LeadCache policy. However, we reserve this task as a future research endeavor. In the revised version, we will make this point more transparent.
>
> **4. On the non-existence of a solution to LP (9) with only one fractional allocation on a cache:** The Reviewer's reasoning is correct. We will explicitly add this point to the proof of Theorem 2.
>
> **5. On defining the fetches formally:** Thanks for pointing this out. We define a fetch event $F(t)$ to take place when the cache configuration at slot $t$ is different from that of the previous slot $t-1$ (i.e., $y_t \neq y_{t-1}$). Hence, when a fetch event happens at a time slot, at least one new file needs to be fetched from the remote server on that slot. In the revised version, we will formally define the “fetch” events.
>
> **6. On the typos:** The reviewer is correct on all three typos. We will correct them in the revised version.
>
> **7. On Societal Impact:** Please see our response (1) above.
>
>
> **References**
>
>
> [1] Kim, Hoon, and Youngnam Han. "A proportional fair scheduling for multicarrier transmission systems." IEEE Communications letters 9, no. 3 (2005): 210-212.
>
>
> [2] Lu, Songwu, Vaduvur Bharghavan, and Rayadurgam Srikant. "Fair scheduling in wireless packet networks." IEEE/ACM Transactions on networking 7, no. 4 (1999): 473-489.
>
> [3] Tassiulas, Leandros, and Saswati Sarkar. "Maxmin fair scheduling in wireless networks." In Proceedings. Twenty-First Annual Joint Conference of the IEEE Computer and Communications Societies, vol. 2, pp. 763-772. IEEE, 2002.
>
> [4] Destounis, Apostolos, Mari Kobayashi, Georgios Paschos, and Asma Ghorbel. "Alpha fair coded caching." In 2017 15th International Symposium on Modeling and Optimization in Mobile, Ad Hoc, and Wireless Networks (WiOpt), pp. 1-8. IEEE, 2017.
>
> [5] Kushner, Harold J., and Philip A. Whiting. "Convergence of proportional-fair sharing algorithms under general conditions." IEEE transactions on wireless communications 3, no. 4 (2004): 1250-1259.
>
> [6] Mukhopadhyay, Samrat, and Abhishek Sinha. "Online Caching with Optimal Switching Regret." arXiv preprint arXiv:2101.07043 (2021).

---

### Official Review · Reviewer_A6cV · 2021-07-17

**Rating:** 5
**Confidence:** 3

**Summary:**

This paper considers an online bipartite caching problem which has a wide application in practice and therefore has been extensively studied. The problem can be expressed via a bipartite graph in which we have a set of vertices on the left, one for each user, and a set of vertices on the right, one for each cache. There is an edge between user vertex and cache vertex if the user can fetch files from the cache. Each cache has a capacity which upper bounds the number of files that it can store. At each time slot, every user may request at most one file to visit. If such a file is stored in one of the neighbors of the user at the current time slot, the objective function will be rewarded by one, otherwise zero. The algorithm needs to decide which files should be stored in which cache at each time slot to maximize the total benefit. The optimization objective is regret which represents the performance of the online algorithm compared to an offline fixed strategy in the hindsight. Upper bounds and lower bounds are shown with respect to regret based on some coloring argument and a previously developed stochastic smoothing framework. The two bounds seem not matching although the authors claim a "tightness".



**Limitations And Societal Impact:**

Yes.

**Main Review:**

The paper studies a caching decision learning problem to minimize regret. Each individual contribution listed in the main results look good but as a whole didn't solve the problem to an extent sufficient for NIPS.

For problems of this nature, one would expect that the upper bound and lower bound of the approximation ratio of the regret match each other. However, although the authors claim in abstract and conclusion that some bound is tight, it cannot be seen clearly in the flow of the paper that this is indeed the case. It would be better for the authors to clarify what that tightness means.

Using transformation from virtual to physical policy looks nice but technically not difficult. Pipage rounding is also kind of standard rounding technique, although using it improves the running of of an earlier work (but the core part inherits that from the earlier work).

Some more detailed comments and typos:

1. It is a bit weird to put the problem formulation in the introduction section which makes the reader almost forget about those definitions when Section 4 Lead Policy starts.

2. Line 44, change "one-hot" to "one-shot".

3. Line 97, change "proceeds" to "proceed".

4. Line 137, change "leads" to "leading".

5. Line 173, delete the first "be".

6. Line 189, delete the last ",".

7. Line 194, this "d" is not defined in the main text but used here.

8. Line 478, a subscript t should be added below the last z.

9. Line 487, delete the redundant "by".

10. Line 521, delete the redundant "the" and change the last "are" to "is".

11. Line 582, change the first "that" to "the".

12. Line 630, delete "of".

13. Line 646, add "are" before "not".



**Time Spent Reviewing:**

18

---

> ### Author Response · Authors · 2021-08-09
> **Thanks for the detailed review**
>
> **1. On the notion of tightness:** We thank the reviewer for pointing out a potential source of confusion. Our results on the regret bounds are tight with respect to their **dependence on the time-horizon** $T$. We prove that the proposed LeadCache policy achieves an $O(\sqrt{T})$ regret and also establish that the regret achieved by any online policy is at least $\Omega(\sqrt{T})$. As we mentioned in the abstract and the discussion following Theorem 5, the constants appearing in the upper and the lower regret bounds differ by a factor of $O(n^{3/8})$. In the revised version, we will make this point more transparent.
>
> **2. On the novelty of the transformation from the physical to the virtual domain:** While we agree with the reviewer that the transformation is not technically difficult, we argue that this transformation is useful and opens up the door of a host of new and non-trivial applications of the FTPL policy to online learning problems with non-linear reward structures. Recall that the FTPL policy is particularly beneficial for structured online learning problems having a combinatorial flavor (e.g., the Online Shortest Path problem). In the following, we give another application of this transformation technique to illustrate its efficacy.
>
>
> **Problem statement:** Consider a caching setting where a single user is connected to a single cache of capacity $C$. However, as opposed to the single-cache problem studied in [1], now assume that the user is interested in simultaneously retrieving **two** separate files of its choice at a round. Thus, the user receives a unit reward if **both** the requested files are stored in the cache or otherwise receive zero rewards. As before, our objective is to design an online caching policy with minimal regret.
>
> **Online Policy:** Let us denote the user's request at the time slot $t$ by an $N \times N$ dimensional symmetric matrix $X^t$, where $X^t_{ij}=X^t_{ji}=1$ if the pair $(i,j)$ was requested by the user at time $t$ and $X^t_{ij}=0$ otherwise. Denoting the set of files cached by the online policy at time $t$ by $y_t$, we can express the one-slot reward function as:
>
> \begin{eqnarray*}
> r_t = \frac{1}{2}\sum_{i, j} X^t_{ij} y_iy_j.
> \end{eqnarray*}
>
> Due to the non-linearity of the reward function, the FTPL policy cannot be employed in this problem directly. To get around this difficulty, we can analogously define an $N\times N$-dimensional symmetric **virtual action** matrix $Z$, such that $Z_{ij} = y_i y_j, \forall i,j$. With this transformation, the slot-wise reward vector takes the standard bilinear form $r_t = Trace(X^t \cdot z_t),$ to which FTPL can be applied. Let us denote the set of all feasible virtual actions as $\mathcal{Z} = \{ Z: Z_{ij} = y_i y_j, \sum_{i} y_i = C, y_i \in \{0,1\} \}.$ The FTPL policy needs to solve a linear optimization problem over the virtual action set $\mathcal{Z}.$ Similar to the problem studied in the paper, it is not difficult to show that this optimization problem is actually **NP-Hard** (Reduction from the Densest $k$-subgraph problem [2]). Let $\Theta_t$ be the cumulative perturbed request count vector (viz. step 6 of Algorithm 1) and let $\mathcal{S}^N_+$ denote the cone of all $N\times N$ real symmetric positive semidefinite matrices.  Mathematically, the FTPL policy at round $t$ needs to solve the following problem:
>
> \begin{eqnarray*}
>  Maximize~ Trace(\Theta_t \cdot Z)
> \end{eqnarray*}
> Subject to,
> \begin{eqnarray*}
> Z &\in& S^N_+,\\\\
> rank(Z) &=& 1,\\\\
> Z_{ii} &\in& \{0,1\}, \forall i \\\\
> Trace(Z) &\leq& C.
> \end{eqnarray*}
> The constraints can be explained as follows: a rank-$1$ positive semidefinite matrix $Z$ can be decomposed as $yy^T$ for some real vector $y.$ The third constraint forces $y$ to be binary. Finally, for binary $y$ vectors, the fourth constraint follows from the cache capacity constraint as shown below:
> \begin{eqnarray*}
> Trace(Z)= \sum_{i}y_i^2 = \sum_i y_i \leq C,
> \end{eqnarray*}
> where we have used the fact that for a binary variable $y \in \{0,1\}$, we have $y^2=y.$
>
> Similar to this paper, one can now suitably relax the non-convex constraints of the above program and attempt to design an $\alpha$-regret policy using low-rank matrix recovery techniques. Please see our response to the reviewer ZWvx in this connection. We will add a summary of the above example in the revised version.
>
> **3. On the typo 2** : Actually, we did mean "one-hot" encoding in the paper. One-hot encoding is a popular technique to vectorize categorical variables [3]. In one-hot encoding, a categorical variable (e.g., a requested file from a user) is represented by an $N$-dimensional vector unit vector that has one in the entry corresponding to the variable's value.
>
> **4. On the typo 7**: We defined the variable $d$ in the Line $19$ of the introduction as the maximum degree of the caches. This variable was also illustrated at the top of Figure 1(b).
>
> **5. On other typos**: We thank the reviewer for a careful reading of the paper. We will correct the typos in the revised version.
>
> **References:**
>
> [1] Bhattacharjee, Rajarshi, Subhankar Banerjee, and Abhishek Sinha. "Fundamental limits on the regret of online network-caching." Proceedings of the ACM on Measurement and Analysis of Computing Systems 4, no. 2 (2020): 1-31.
>
> [2] Sotirov, Renata. "On solving the densest k-subgraph problem on large graphs." Optimization Methods and Software 35, no. 6 (2020): 1160-1178.
>
> [3] Rodríguez, Pau, Miguel A. Bautista, Jordi Gonzalez, and Sergio Escalera. "Beyond one-hot encoding: Lower dimensional target embedding." Image and Vision Computing 75 (2018): 21-31.

---

### Official Review · Reviewer_ZWvx · 2021-08-01

**Rating:** 8
**Confidence:** 3

**Summary:**

This paper proposed an algorithm for the bipartite caching problem. The proposed algorithm has a follow-the-perturbed-leader (FTPL) flavour. The regret under the policy is proved to be optimal up to constant factors, assuming an adversarial request sequence. Further, it is proved if the request sequence is i.i.d. (or more generally satisfies a law of large numbers type property), then the cache stabilizes in finite time with probability 1.

**Main Review:**

The results are quite strong. The novelty seems to lie in being able to relax the non-linear problem into a linear one, allowing FTPL regret bounded techniques to apply.

One aspect that I think deserves clarification in the manuscript is the following: The regret guarantee is proved assuming the optimal placement corresponding to Line 7 of Algorithm 1, which is NP-hard. While the authors do establish an approximation guarantee for a rounding technique, it is not clear how using an approximate solution of (7) would impact the regret of Algorithm 1. The authors may not have a clean answer to this, but this `gap' definitely deserves a discussion in the paper.

However, in spite of the above, I think the paper makes a notable contribution.

**Update: I thank the authors for the elaborate and comprehensive response. It would indeed be great to introduce the result on \alpha-regret in the paper (if accepted).

**Time Spent Reviewing:**

3

---

> ### Author Response · Authors · 2021-08-09
> **An efficient randomized rounding scheme with a sublinear $(1-1/e)$-regret**
>
> We thank the reviewer for raising this important point. Indeed we can show that the LeadCache policy, with an efficient randomized rounding algorithm described below, provably achieves a sublinear $(1-1/e)$-regret. Before describing our result, we give a brief background on the notion of $\alpha$-regret.
>
> **1. Background**
>
> Since the optimization problem in Line 7, Algorithm 1 is **NP-Hard**, it is natural to search for an **efficient** online policy with a sub-linear $\alpha$-regret, denoted by $R_T^\alpha$, for some constant $0<\alpha \leq 1$. Note that the quantity $\alpha$-regret is defined analogously to the usual static regret where the reward accrued by the optimal offline policy is discounted by a factor of $\alpha$ [1], i.e.,
> \begin{eqnarray*}
> R_{\pi, T}^\alpha = \sup_{\{x_t\}} \bigg(\alpha \sum_{t=1}^T q(x_t, y^*) - \sum_{t=1}^T q(x_t, y_t^\pi)\bigg),
>  \end{eqnarray*}
> with the usual meaning of the symbols as defined in Section 1.1. Directly using the Pipage rounding scheme described in Algorithm (2) does not necessarily yield an online policy with a sub-linear $\alpha$-regret. In fact, the problem of designing an efficient sub-linear $\alpha$-regret policy for a problem with a given $\alpha$-approximation offline oracle is non-trivial. In the paper [2], Kakade et al. proposed an offline-to-online reduction that makes a **linearly increasing** number of per-iteration calls to the offline approximation oracle. See also the offline-to-online reduction schemes given in the papers [3], [4], and [5]. However, all of these generic reductions need to make multiple calls to the approximation oracle per round, and hence, do not scale well. For the LeadCache problem, we now give a very efficient offline-to-online reduction that makes only a **single** call per round to a linear-time randomized approximation oracle. For this, we recall the following definition from [1].
>
> **Definition** ($\alpha$ point-wise approximation):
> 	 For an integral feasible set $\mathcal{Z}$ in the non-negative orthant and a non-negative input vector $x,$ consider the Integer Linear Program $\max_{z \in \mathcal{Z}}  z \cdot x.$ Let $\mathcal{Z}' \supseteq \mathcal{Z}$ be a relaxation of the feasible set $\mathcal{Z}$ and let $z \in \arg\max_{z \in \mathcal{Z}'} z \cdot x$ be an optimal solution of the relaxed ILP. If for some $0< \alpha \leq 1,$ a (randomized) rounding algorithm $A$ returns a feasible solution $\hat{z} \in \mathcal{Z}$ such that $ \mathbb{E}\hat{z}_i \geq \alpha z_i, \forall i,$ and for any input $x$, we call the algorithm $A$ an $\alpha$-**point-wise approximation**.
>
> It immediately follows that for any $\alpha$ point-wise approximation algorithm for the optimization problem in Line 7, Algorithm 1, if $z_t \in \mathcal{Z}'$ is the relaxed virtual action at time $t$, we have
> \begin{eqnarray*}
> \sum \mathbb{E}(\hat{z}_t) \cdot x_t  \geq \alpha \sum z_t \cdot x_t,
> \end{eqnarray*}
> where the inequality follows from the point-wise approximation guarantee. Thus, for any $z^* \in \mathcal{Z},$ the $\alpha$-regret of a  virtual policy can be bounded as:
> \begin{eqnarray*}
> 	\alpha \sum x_t \cdot z^* - \sum \mathbb{E}(\hat{z}_t) \cdot x_t \leq \alpha (\sum x_t \cdot z^* - \sum x_t \cdot z_t  \big) \leq \alpha \tilde{R}_T^{LeadCache},
> \end{eqnarray*}
>  where $\tilde{R}_T^{LeadCache}$ is the regret of the LeadCache policy with the relaxed set of actions $\mathcal{Z}'$. The regret $\tilde{R}_T^{LeadCache}$ can be bounded by $O(n^{3/4} \sqrt{dmCT})$ by following the same line of arguments as in the proof of Theorem 1 (see a proof-sketch in the Appendix below). Hence, we only need to design an $\alpha$ point-wise approximation scheme for the offline optimization problem to ensure a sublinear $\alpha$-regret.
>
> Our main ingredient for designing the $\alpha$ point-wise approximation scheme is Madow's systematic sampling scheme, commonly used in the statistical sampling literature [6]. Given a set of inclusion probabilities **p** on a set of items $[N]$, Madow's scheme efficiently outputs a subset $S \subseteq [N]$ of size $C$ in **linear time** such that the $i$th element is included in the subset $S$ (without replacement) precisely with probability $p_i, 1\leq i \leq N.$ For this sampling scheme to work, it is necessary and sufficient that the inclusion probability vector **p** satisfies the following feasibility constraint [6]:
> \begin{eqnarray*}
> 	\sum_{i=1}^N p_i =C, \textrm{ and }
> 	0\leq p_i \leq 1, \forall i \in [N].
> \end{eqnarray*}
> With this background, we now proceed to design an $\alpha$ point-wise approximation scheme for the **NP-hard** optimization problem in Line 7, Algorithm 1.
>
>
> **2. Design of an efficient $(1-1/e)$-Regret Caching Policy for LeadCache:**
>
> Our $\alpha$ point-wise approximate rounding scheme independently samples $C$ files in each cache $j$ according to the inclusion probabilities given by the vector $y^j,$ which is obtained from the solution of the  relaxed LP (9) at each round. From the constraints of the LP, it follows that each of the vectors $y^j$'s satisfies the above feasibility constraint, and hence, Madow's sampling scheme applies.
> To show that the resulting rounding scheme satisfies the $\alpha$ point-wise approximation property, note that for all $i \in \mathcal{I}$ and $f \in [N]:$
> \begin{eqnarray*}
> \mathbb{P}(\hat{z}^i_f =1) = \mathbb{P}( \bigvee_{j \in \partial^+(i)} \hat{y}^j_f =1)
>  \stackrel{(a)}{=} 1- \prod_{j \in \partial^+(i)}(1-y^j_f)
>  \stackrel{(b)}{\geq} 1 - e^{-\sum_{j \in \partial^+(i)} y^j_f }
>  \stackrel{(c)}{\geq} 1 - e^{-z^i_f}
>  \stackrel{(d)}{\geq} (1-\frac{1}{e})z^i_f,
> \end{eqnarray*}
> where (a) follows from the fact that rounding in each caches are done independently of each other, (b) is a standard algebraic inequality $e^x \geq 1+x, \forall x \in \mathbb{R}$, (c) follows from the feasibility constraints of the LP, and (d) follows from the concavity of the function $1-\exp(-x)$ and the fact that $0\leq z^i_f \leq 1.$ Hence, the LeadCache policy with the above randomized rounding scheme achieves a sub-linear $\alpha = 1-1/e$ regret.
>
> However, it is not clear at this point whether the Pipage rounding scheme also enjoys a similar $\alpha$-regret guarantee. We will add the above result in the revised version of the paper.
>
> **3. Appendix:**
>
> **Proposition $\tilde{R}_T^{LeadCache} = O(n^{3/4} \sqrt{dmCT})$**
>
> **Proof Sketch:** The proof of the regret bound with the relaxed actions follows exactly the same line of arguments as the proof of Theorem 1. In particular, we decompose the regret bound as in Eqn (26) of supplement with the feasible set $\mathcal{Z}$ in term (b) replaced by the relaxed feasible set $\mathcal{Z}'$.  Steps for bounding the term (c) remains unchanged. However, for bounding the Gaussian complexity in term (b), we had explicitly used the fact that the cache allocations are integral (viz. the counting argument in Eqn. (28)). This argument does not apply to the relaxed set $\mathcal{Z}'.$ For this, we now give a different argument for bounding the Gaussian complexity in term (b) with the relaxed action set $\mathcal{Z}'.$ Note that for any feasible $(z, y) \in \mathcal{Z}',$ we can bound the $1$-norm of $z$ as follows:
> \begin{eqnarray*}
>  ||z||^1 = \sum_{i,f} z^i_f \leq d\sum_{j}\sum_{f} y_{f}^j \leq mCd,
> \end{eqnarray*}
> where we have used the degree bounds and the cache capacity constraints. Hence, for a set of $Nn$ i.i.d. standard Gaussian random variables we have
> \begin{eqnarray*}
> \mathcal{G}(\mathcal{Z}') = \mathbb{E}(\max_{(z \in \mathcal{Z}')} \langle z,\gamma \rangle) \stackrel{Holder's~ ineq.}{\leq} \mathbb{E}(\max_{ (z \in \mathcal{Z}')} || z||^{1} ||\gamma ||^{\infty}) \stackrel{(a)}{\leq} mCd \sqrt{4 \ln(Nn)},
> \end{eqnarray*}
> where in the inequality (a), we have used a standard bound on the expectation of the maximum of the absolute value of a set of i.i.d. standard Gaussian random variables (Massart's lemma) with the above bound. Now proceeding similarly as in the proof of the regret bound for the action set $\mathcal{Z}$, we get the regret bound as claimed.
>
>
>
> **References:**
>
> [1]. Kalai, Adam, and Santosh Vempala. "Efficient algorithms for online decision problems." Journal of Computer and System Sciences 71, no. 3 (2005): 291-307.
>
> [2]. Kakade, Sham M., Adam Tauman Kalai, and Katrina Ligett. "Playing games with approximation algorithms." SIAM Journal on Computing 39, no. 3 (2009): 1088-1106.
>
> [3]. Garber, Dan. "Efficient Online Linear Optimization with Approximation Algorithms." Mathematics of Operations Research 46, no. 1 (2021): 204-220.
>
> [4]. Hazan, Elad and Hu, Wei and Li, Yuanzhi and Li, Zhiyuan. "Online Improper Learning with an Approximation Oracle." Advances in Neural Information Processing Systems 2018, vol 31.
>
> [5] Fujita, Takahiro, Kohei Hatano, and Eiji Takimoto. "Combinatorial online prediction via metarounding." In International Conference on Algorithmic Learning Theory, pp. 68-82. Springer, Berlin, Heidelberg, 2013.
>
> [6] Madow, William G. "On the theory of systematic sampling, II." The Annals of Mathematical Statistics 20, no. 3 (1949): 333-354.

---

### Decision · Program_Chairs · 2021-09-27

**Decision:**

Accept (Poster)

**Comment:**

Overall, this is a quality submission. The paper addresses an important problem of interest to the community. They were given new algorithmic and analysis insights into bounding regret for this problem.  The reviewers felt the paper was written well and would be of interest.